# SCIATRAN software package (V4.6): update and further development of aerosol, clouds, surface reflectance databases and models

**Linlu Mei, Vladimir Rozanov, Alexei Rozanov, and John P. Burrows**

Institute of Environmental Physics, University of Bremen, Bremen, Germany

**Correspondence:** Linlu Mei (mei@iup.physik.uni-bremen.de)

**Abstract.** Since the initiation of development at the Institute of Environmental Physics (IUP), University of Bremen, in 1994, the radiative transfer model SCIATRAN (formerly GOMETRAN) has been continuously improved and new versions have been released (Rozanov et al., 1997, 2002, 2005, 2014, 2017). In the course of development, the SCIATRAN software package became capable of simulating radiative transfer processes through the Earth's atmosphere or coupled atmosphere–ocean system with a variety of approaches to treat the sphericity of the atmosphere (plane-parallel, pseudo-spherical, approximately spherical and full-spherical solutions) in both scalar and vector modes. Supported by a variety of built-in databases and parameterizations, these capabilities made SCIATRAN widely used for various remote-sensing applications related to the retrieval of atmospheric trace gases and characteristics of aerosols, clouds and surfaces. This paper presents an overview of the cloud, aerosol and surface (CAS) databases and models implemented in the SCIATRAN software package (V4.6) and provides some recommendations on their usage. The new implementations offer potential users a flexible interface to perform radiative transfer simulations: (1) accounting for multi-layer liquid water, ice and mixed-phase clouds; (2) employing typical aerosol-type parameterizations (including vertical variability) used in the satellite and model communities as well as updated databases; (3) including various surface bidirectional reflectance distribution function (BRDF) and albedo models for land, vegetation, ocean, snow and melt ponds on sea ice.

The most recent version of the radiative transfer model SCIATRAN is freely available at the website of the IUP, University of Bremen: http://www.iup.physik.uni-bremen.de/sciatran (last access: November 2022).

## 1 Introduction

Shortly after the beginning of the Space Age in 1957, the first Earth observation satellite flew around the Earth and discovered the Van Allen radiation belts. Since then, Earth observation satellites have been evolving, and a global observation system utilizing remote-sensing instrumentation is now being developed. The satellite instruments have been continually evolving in the past decades. However, the explosion in the exploration of Earth observation data would not have been possible without (i) the simultaneous and remarkable increase in the speed of computers and their data storage and (ii) the development of sophisticated forward radiative transfer models (RTMs) and retrieval algorithms.

Increasingly accurate RTMs have been developed to meet the need to simulate Earth observation data from the evolving observing system. SCIATRAN, which is the subject of this paper, is a comprehensive radiative transfer software package. SCIATRAN can simulate radiative transfer processes, including polarization through Earth's atmosphere–ocean–land–cryosphere system in the spectral range from 175 nm to 40 µm for any observation/illumination geometries (Rozanov et al., 2014).

During the last decades, SCIATRAN has become a well-known and widely used radiative transfer model. The most important application area for the SCIATRAN user community is the retrieval of atmospheric and surface parameters

using remote-sensing data. Figure 1 shows the global distribution of SCIATRAN users as of November 2022. One can see that the SCIATRAN software has more than 1000 external users from 315 cities in 58 countries. The SCIATRAN software package has been cited by about 1400 publications (source: Google Scholar, https://scholar.google.com/, last access: November 2022).

Continuous developments of the SCIATRAN software include two aspects: (i) implementation of new features in the radiative transfer and retrieval modules and (ii) update and implementation of databases and new surface reflectance models. This paper focuses on the update of cloud and aerosol databases and the implementation of new surface bidirectional reflectance distribution function (BRDF) models in the most recent SCIATRAN version – V4.6 (Rozanov et al., 2022). A comprehensive overview of the existing cloud, aerosol and surface (CAS) databases and models was presented by Rozanov et al. (2014). However, a major extension of the CAS databases and models in SCIATRAN has been done since 2014. To bring the user community up to date with the options available within SCIATRAN, this paper describes the improved cloud and aerosol databases and the usage of new surface BRDF models available in SCIATRAN.

The paper is structured as follows. Section 2 contains brief information about the general capabilities of the SCIATRAN software. Section 3 presents a brief summary of the new features in SCIATRAN V4.6. In Sect. 4.1 we summarize the new CAS databases and models implemented in SCIATRAN. Section 5 describes the optical properties of the selected CAS. Section 6 shows some examples and comparisons of forward simulations using the new implementations. The conclusions are presented in Sect. 7.

## 2  General capabilities of the SCIATRAN software

This section gives an overview of the main features and general capabilities of the SCIATRAN software, which can be of interest to potential users.

Using SCIATRAN, polarized (vector) and scalar radiative transfer calculations can be performed in plane-parallel, pseudo-spherical, approximately spherical or fully spherical geometry. Vertical inhomogeneities for parameters such as temperature, pressure, gaseous absorber concentration, aerosol particle number density and multilayer liquid water and ice clouds can be easily accounted for. For inexperienced users, the option of a vertically homogeneous medium, which requires a minimum number of input parameters, might be useful.

The surface reflection can be described as (1) wavelength-dependent Lambertian reflection or constant albedo or (2) BRDF including parameterizations for a variety of surface types (ocean, vegetation, soil, snow, melt ponds and sea ice). In particular, in the case of light reflection by an ocean surface, the SCIATRAN software enables us to consider the Fresnel reflection from absolutely calm and wind-roughed surfaces (accounting for polarization), whitecap reflection and water-leaving radiation. Reflection by white ice and melt ponds on sea ice can also be accounted for. In addition, a coupled ocean–ice–atmosphere system can be considered to simulate the radiative transfer over and under the ice or water surface.

The software package has been designed to perform fast and accurate simulations of radiance spectra appropriate to atmospheric remote-sensing observations in the ultraviolet (UV)–visible (VIS)–near infrared (NIR)–shortwave infrared (SWIR)–thermal infrared (TIR) spectral ranges. Different models and databases included in SCIATRAN are valid in different spectral ranges. In the spectral range 225–2500 nm, most of the described database can be used. Generally, the calculations can be performed in the spectral range from 175.44 nm to 40 μm, including $O_2$ Schumann–Runge and Herzberg absorption bands in the UV spectral range, various gaseous absorbers such as $O_3$, $NO_2$, CIO, OCIO, BrO, HCHO, $SO_2$, $NO_3$, $O_4$, $O_2$, $H_2O$, $CO_2$, CO, $CH_4$ and $N_2O$ throughout the whole spectral range, and thermal emission in the TIR spectral range. Depending on the spectral range, some limitations in the functionality might apply.

All relevant elastic scattering processes such as the Rayleigh scattering, aerosol and cloud scattering and absorption can be taken into account. Additionally, important inelastic scattering processes such as vibrational Raman scattering and colored dissolved organic matter (CDOM) fluorescence in the ocean (Rozanov et al., 2017; Wolanin et al., 2015) and rotational Raman scattering in the atmosphere (including electron spin-rotational splitting) (Rozanov and Vountas, 2013; Lelli et al., 2017; Rozanov et al., 2021) can be accounted for. SCIATRAN also includes a retrieval block, which enables a continually increasing number of Earth observation data products from ground-based, ship, aircraft or satellite-borne sensors to be delivered in near real time. This block, however, is not a part of the freely distributed software package.

Employing the SCIATRAN software, users can calculate the radiance/intensity (scalar mode) or Stokes vector (vector mode), scalar or vector-weighting functions (Jacobian matrix) for concentrations of gaseous absorbers, cloud optical thickness, cloud top and bottom heights, effective radius of water droplets and ice crystals, aerosol number density, surface albedo and many other parameters,[1] such as air mass factors (AMFs), slant columns, vertically resolved AMFs (box AMFs), fluxes (actinic, upwelling, downwelling, diffuse and total), spherical albedo, diffuse transmission and vertical optical depth. AMFs of a radiation field such as fluxes, spherical albedo and diffuse transmission can also be calcu-

---

[1]A complete list of all available parameters can be found in the SCIATRAN user guide.

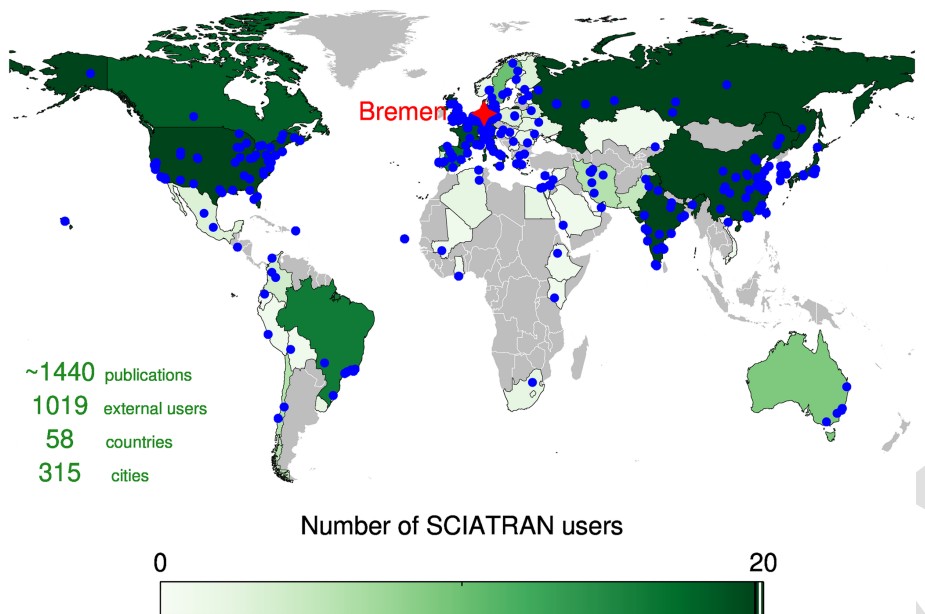

**Figure 1.** Geographic distribution of SCIATRAN users worldwide as of November 2022. The color indicates the total number of users in each particular country, and the blue dot is the city where SCIATRAN users are located. The red star indicates the city of Bremen.

lated assuming specific isotropic illumination conditions of a medium from above and below.

The SCIATRAN software contains not only different solvers (discrete-ordinates method (DOM) (Siewert, 2000), combined differential-integral (CDI) approach (Rozanov et al., 2000) initialized by either DOM or finite-difference method (FDM) (Barkstrom, 1976), and the CDI approach involving the Picard iterative approximation (Rozanov et al., 2001) initialized by either DOM or FDM solution), which are capable of performing radiative transfer calculations through a plane-parallel or spherical atmosphere but also numerous parameterizations of the Rayleigh scattering, databases of aerosol and cloud optical properties as well as various models of the surface BRDF, which significantly facilitates the usage of the software, especially for novice users. Taking into account that parameterizations of the Rayleigh scattering have been discussed in Rozanov et al. (2014), the following sections will present newly implemented databases of aerosol and cloud optical properties as well as models of the bidirectional surface reflection. A consideration of databases for numerous gaseous absorbers, including the HITRAN 2020 molecular spectroscopic database (Gordon et al., 2022), is beyond the scope of this paper.

## 3 Overview of new SCIATRAN features

While the initial development of SCIATRAN was dedicated to the satellite missions GOME (Global Ozone Monitoring Experiment) and SCIAMACHY (SCanning Imaging Absorption spectroMeter for Atmospheric CHartogra-

phy) and their products (Burrows et al., 1995; Bovensmann et al., 2010; Buchwitz et al., 2013; Richter et al., 2005; Noël et al., 2018; Arosio et al., 2018; Weber et al., 2018), the implementation of CAS databases and models in SCIATRAN is mainly driven by the developments of the eXtensible Bremen Aerosol/cloud and surfacE parameters Retrieval (XBAER) algorithm and its products. The XBAER algorithm is now capable of providing global data products for cloud (Mei et al., 2017b, 2018a, 2019), aerosol (Mei et al., 2017a, 2018b, 2020a, b) and surface parameters (Mei et al., 2021a, b). The further developments of SCIATRAN support (1) the design of new instruments (e.g., Carbon Monitoring Satellite (CarbonSat), Buchwitz et al., 2013, Methane Airborne MAPper (MAMAP) Gerilowski et al., 2011), (2) the release of new versions of atmospheric and surface satellite products at the Institute of Environmental Physics (IUP), University of Bremen, and (3) the exploration of new applications for related research topics worldwide.

Table 1 summarizes new implementations of CAS since the paper of Rozanov et al. (2014). All CAS databases and models from the previous versions of SCIATRAN are still available in V4.6. In particular, (1) cloud databases include liquid water clouds with an effective radius of water droplets in the range 4–20 µm and ice clouds consisting of ice crystals with the shape of a second-generation fractal. (2) Aerosol databases include OPAC version 3, World Meteorological Organization (WMO) and Low-Resolution Atmospheric Transmission (LOWTRAN). (3) Models for surface BRDF are RPV, modified RPV and RossThick-LiSparseReciprocal (RTLSR). In V4.6, optical characteristics of the liquid water cloud database have been extended to cover the water droplet

effective radii between 2 and 40 μm, in contrast to 4–20 μm in the previous version. The Baum (Baum et al., 2011) and Yang (Yang et al., 2013) ice crystal databases have been included in SCIATRAN. Many important details of the Yang database implementation can be found in Pohl et al. (2020).

Aerosol-type parameterizations such as the Moderate Resolution Imaging Spectroradiometer (MODIS) Dark Target (DT) over land (Levy et al., 2013) and ocean (Remer et al., 2005), the Modern-Era Retrospective analysis for Research and Applications (MERRA) (Randles et al., 2017) and XBAER (Mei et al., 2017a) have been implemented. In addition, the updated Optical Properties of Aerosols and Clouds (OPAC) dataset (Koepke et al., 2015) and optical parameters of dust aerosol particles according to Dubovik et al. (2006) have also been implemented.

The XBAER surface reflectance parameterization (Mei et al., 2017a), RossThick-LiSparseReciprocal-Snow (RTL-SRS) (Jiao et al., 2019), Fast and Accurate Semi-analytical Model of Atmosphere-surface Reflectance (FASMAR) (Mei et al., 2022) and BRDF models for snow, white ice and melt ponds on sea ice are now available.

In recent years, to address the exponentially increasing amount of remote-sensing data, fast radiative transfer models focusing on specific data products have been developed. Examples are that (i) FASMAR (Mei et al., 2022) is used with XBAER to retrieve surface (e.g., snow grain size, specific surface area) and aerosol products (e.g., aerosol optical thickness – AOT) and that (ii) FOCAL (Fast atmOspheric traCe gAs retrievaL) (Reuter et al., 2017) uses a simplified RTM to determine the dry mole fractions of carbon dioxide, $XCO_2$, and methane, $XCH_4$, from measurements of the radiance in the NIR and SWIR. These models have also benefited from SCIATRAN, in its being a high-accuracy reference RTM.

## 4 CAS databases and models implemented in SCIATRAN

In this section, we briefly introduce the CAS databases and models newly implemented in SCIATRAN. Almost all the databases can be used to perform radiative transfer simulations in the spectral range at least between 225 nm and 2.5 μm.

### 4.1 Aerosol

In our seminal paper (Rozanov et al., 2014) dedicated to the software package SCIATRAN, such aerosol models and databases as WMO (Bolle, 1986), LOWTRAN (Levoni et al., 1997) and OPAC version 3 (Hess et al., 1998) were described. We recall that the WMO and LOWTRAN databases contain optical characteristics (extinction coefficient, single-scattering albedo and scattering matrix – or phase function in the scalar case) of different aerosol components and aerosol types, respectively. In particular, in the WMO database six

aerosol components are included, i.e., water-soluble, dust, oceanic, soot, stratospheric and volcanic.

The commonly used aerosol types (e.g., continental, maritime, urban) are defined by the external mixture of the optical properties of those six components. The user has to specify the vertical profile of the aerosol extinction coefficient at a reference wavelength.

The LOWTRAN database (version 7) consists of extinction and scattering coefficients and asymmetry parameters for predefined aerosol types located in different atmospheric layers (i.e., boundary layer, troposphere, stratosphere and mesosphere). The basic aerosol types in the boundary layer include rural, urban and maritime aerosols. Only one type is available for the tropospheric aerosols. For both the boundary layer and the troposphere, the humidity and visibility are to be selected by the users. In the stratosphere, users can select (1) either background or volcanic (moderate, high, extreme) aerosol loading or (2) either a fresh or aged volcanic aerosol type. For the mesosphere a choice between a normal aerosol state and a transition from the volcanic stratosphere to the normal mesosphere is available.

The OPAC (version 3) database (Hess et al., 1998) implemented in the SCIATRAN software consists of refractive indices and parameters of lognormal particle size distribution functions of predefined components such as insoluble, meteoritic dust, mineral (nuc., acc., coa., mode), mineral-transported, soot, sulfate, sea salt (acc., coa., mode) and water-soluble. The optical properties of an aerosol type are calculated as the external mixture of predefined aerosol components using the Mie code published by Mishchenko et al. (1999), which is incorporated into SCIATRAN. The calculation of optical characteristics is performed automatically by the SCIATRAN software without any further actions by users if option OPAC is selected.

The following new aerosol databases have been implemented in the recent SCIATRAN version: an updated OPAC database, MODIS Dark Target (both land and ocean), MERRA, XBAER and dust aerosol types. These aerosol types provide aerosol parameterization methods, which are well-received by the aerosol community.

When selecting a suitable aerosol database or parameterization, besides the requirement with respect to the appropriate spectral range, a possibility of using altitude-dependent particle number density and optical properties (extinction cross section, single-scattering albedo (SSA) and phase function) is also an important issue. In contrast to the aerosol particle number density, which can be defined as a function of altitude, for all newly implemented aerosol types in SCIATRAN, altitude-dependent particle optical properties can only be defined when using the MERRA or OPAC aerosol types. We note that SCIATRAN users can define a specific altitude-dependent aerosol type using OPAC microphysical properties. However, the usage of MERRA types is significantly simpler as compared to OPAC, because it does not require from the user any a priori knowledge about

**Table 1.** Databases and models described in Rozanov et al. (2014) and in this paper.

| Paper | Rozanov et al. (2014) | This paper |
|---|---|---|
| Cloud | Liquid water and ice clouds | Update of liquid water and ice clouds |
| Aerosol | OPAC 3, WMO, LOWTRAN | OPAC 4, MERRA, XBAER, MODIS Dark Target, spheroidal dust |
| Surface | RPV, modified RPV (mRPV), mRPV plus facet, RTLSR | XBAER, FASMAR, RTLSRS, snow and melt ponds on sea ice |

the chemical composition and microphysical properties of aerosol types in different atmospheric layers. The aerosol types which were available in previous SCIATRAN versions (e.g., WMO and LOWTRAN) enable the users to describe the altitude-dependent optical properties only for predefined altitude layers (boundary layer, troposphere, stratosphere and mesosphere). The newly implemented MODIS Dark Target, XBAER and dust aerosol parameterizations define altitude-independent aerosol types.

### 4.1.1 OPAC aerosol types

In the previous version of the SCIATRAN paper (Rozanov et al., 2014), the details of the OPAC version 3 database (Hess et al., 1998) were described.

A new version of OPAC (version 4.0) was released recently by Koepke et al. (2015). The main improvement is the update of the shape of mineral dust, from spherical to prolate spheroids. The usage of a spherical assumption for dust particles was proven to introduce large errors for both aerosol property retrieval (Zhao et al., 2003) and dust radiative forcing calculations (Mishchenko, 2001). The usage of the OPAC (version 4.0) database within the SCIATRAN software does not require Mie calculations because the database includes phase functions at a discrete number of scattering angles and extinction and scattering coefficients precalculated by employing the T-matrix and geometric optics methods (Koepke et al., 2015).

### 4.1.2 MODIS Dark Target (MODIS-DT) aerosol types

The MODIS-DT aerosol type is operationally used to derive the NASA aerosol property dataset from MODIS and Visible Infrared Imaging Radiometer Suite (VIIRS) instruments (Levy et al., 2013; Remer et al., 2020). This aerosol type provides an aerosol parameterization method, which is well-received by the aerosol community.

The MODIS Dark Target aerosol-type parameterization is different for aerosols over the land and ocean.

**Dark Target aerosol type over land (MODIS-DT)**

The MODIS-DT aerosol parameterization implemented in the SCIATRAN software offers the following aerosol types: weakly absorbing (WA), moderately absorbing (MA), strongly absorbing (SA) and specific spheroidal (dust). Although the continental aerosol type was also included in the original MODIS Dark Target over land aerosol parameterization, this type is not included in the current SCIATRAN database because (1) the continental aerosol type is used only when none of the three aerosol types (WA, MA and SA) can provide satisfactory retrieval results in the Dark Target algorithm for the MODIS instrument (personal communication with Dr. Robert Levy) and (2) this type is not used in the XBAER algorithm as it was found by us to provide unsatisfactory results for the MEdium Resolution Imaging Spectrometer (MERIS) instrument.

Each aerosol type contains fine and coarse modes (hereafter Modes 1 and 2), which are characterized by the volume particle size distribution (PSD) given by

$$\frac{dV}{d\ln r} = \frac{V_0}{\sqrt{2\pi}\,\sigma} \exp\left[-\frac{(\ln r - \ln r_v)^2}{2\sigma^2}\right], \tag{1}$$

where $V_0$ is the total volume of particles in a vertical column of a unit area, $r_v$ is the mode radius and $\sigma$ is the standard deviation. Parameters of the aerosol PSD function and the refractive index (RI) for each mode are represented as functions of the AOT at a wavelength of 0.55 μm. The parameterization formulas for each aerosol type and mode are listed in Table 2. We note that the relationships presented in the table are valid for $\tau \le 2$ for the MA and SA types and for $\tau \le 1$ for the WA and dust types. For larger values of $\tau$, $\tau = 2$ and $\tau = 1$, respectively, are to be used in the formulas.

For a selected aerosol type and AOT, SCIATRAN calculates parameters of the volume PSD and RI for the fine and coarse modes according to Table 2. Then, employing incorporated Mie code, the scattering and absorption cross sections as well as the scattering matrix are calculated. The optical parameters for the selected aerosol type are calculated thereafter as follows:

$$\sigma_P = f\,\sigma_{p_1} + (1-f)\,\sigma_{p_2}, \tag{2}$$

where subscript "p" denotes a scattering "s", absorption "a" or extinction "e" cross section and $f$ is the fraction of the fine mode given by $f = N_{01}/(N_{01} + N_{02})$. To calculate the fine-mode fraction, the following relationship between the total

volume, $V_0$, and the total particle number in a vertical column of a unit area, $N_0$, is employed:

$$N_0 = V_0 \frac{3}{4\pi r_{\mathrm{v}}^3} \exp\left(\frac{9\sigma^2}{2}\right) . \tag{3}$$

The phase function is calculated as

$$p(\gamma) = \frac{\sigma_{s_1} p_1(\gamma) + \sigma_{s_2} p_2(\gamma)}{\sigma_{\mathrm{s}}} , \tag{4}$$

where $\gamma$ is the scattering angle.

The usage of MODIS-DT aerosol-type parameterization within the SCIATRAN software requires the user to select one of the above-discussed aerosol types and to provide AOT at a wavelength of 0.55 μm. In addition, the user can define any shape, $S_{\mathrm{h}}(z)$, of the aerosol number density vertical distribution (the default shape is a constant within an aerosol layer). The number density profile is then calculated as

$$N(z) = \frac{\tau}{\sigma_{\mathrm{e}} \int\limits_0^H S_{\mathrm{h}}(z)\,\mathrm{d}z} S_{\mathrm{h}}(z) , \tag{5}$$

where $H$ is the top height of the aerosol layer. In the operational MODIS-DT aerosol product, the aerosol layer is assumed to be located between 0 and 3 km with an exponential number density vertical distribution.

MODIS-DT aerosol types were used in the framework of the SCIATRAN software to calculate the lookup tables widely used in the AOT retrieval algorithm over land (Mei et al., 2017a, 2020b).

**Dark Target aerosol type over the ocean (MODIS-OC)**

Unlike the land case, the aerosol type over the ocean in the DT algorithm is defined by selecting one of the four fine modes (F1, F2, F3, F4) and one of the five coarse modes (C1, C2, C3, C4, C5). The PSD parameters and RI are given in Table 3. For wavelengths selected by users, the Mie code is used by the SCIATRAN software to calculate scattering and extinction cross sections as well as scattering matrices. The fine mode, coarse mode and fraction of coarse mode are input parameters. In contrast to the MODIS-DT aerosol type, which requires AOT as an input parameter, a vertical profile of the aerosol particle number density in physical units needs to be provided over the ocean. Using the calculated extinction cross section and the number density profile, AOT is calculated and written out by the SCIATRAN software.

### 4.1.3 XBAER aerosol type over the ocean (XBAER-OC)

The XBAER aerosol type was originally designed for usage within the framework of the XBAER retrieval algorithm. A similar strategy, as described by Levy et al. (2007), has been employed to obtain aerosol types over the ocean. In particular, the statistical analysis was performed using the Maritime

Aerosol Network (MAN) observations (Smirnov et al., 2009) and four aerosol types; i.e., pure maritime, pollution 1, pollution 2 and dust-influenced were defined. Table 4 shows the volume PSD parameters and RI of these predefined aerosol types as used in the XBAER algorithm over the ocean and implemented in the SCIATRAN software.

A slight difference as compared to the MODIS-DT aerosol types is that the XBAER ocean aerosol type is parameterized not only with respect to AOT at 0.55 μm, $\tau$, but also with respect to fine-mode AOT at 0.55 μm, $\tau_f$. The fine-mode AOT is defined as the product of the total AOT at 0.55 μm and the fine-mode volume fraction, which is also a function of $\tau$. The fine-mode volume fraction is calculated as $f_{\mathrm{v}} = V_{01}/(V_{01} + V_{02})$, where $V_{01}$ and $V_{02}$ are given in Table 4 as a function of $\tau$ for Modes 1 and 2, respectively.

The input parameters for this parameterization are the aerosol type (pure maritime, pollution 1, pollution 2 or dust-influenced), AOT at 0.55 μm and the shape of the aerosol number density profile, similarly to the MODIS-DT aerosol type over land.

A combination of MODIS-DT for land and XBAER-OC for ocean aerosol types enables the same treatment of global aerosol parameterization. For a given location and time, a unique aerosol type can be selected and used for global aerosol retrieval over land and ocean (Mei et al., 2017a, 2018a). The XBAER aerosol type is operationally used to derive the ESA Climate Change Initiative (CCI) and Copernicus Climate Change Service (C3S) aerosol property dataset from the MERIS and Ocean and Land Colour Instrument (OLCI) instruments using a lookup table calculated with the SCIATRAN software (Mei et al., 2017a, 2018b).

### 4.1.4 Dubovik dust aerosol type

Dust aerosol is a very important aerosol type. Many investigations have been done specifically for dust aerosol. For example, compared to OPAC version 3.0, OPAC version 4.0 mainly improves the treatment of the dust. In the MODIS-DT aerosol type, the spheroidal dust can also be used. However, the online calculation of the aerosol optical parameters in the case of nonspherical particles is usually very time-consuming. To reduce the calculation time, the precalculated kernel lookup tables and software to calculate optical parameters of the dust aerosol type were presented by Dubovik et al. (2006). This software package is not yet included in SCIATRAN, but SCIATRAN is capable of reading optical parameters of dust aerosol calculated externally by employing the kernel lookup table. Thus, the input parameters for this aerosol type are precalculated optical parameters, AOT at selected wavelengths and the aerosol number density profile.

The Dubovik dust aerosol type is operationally used to derive the ESA CCI aerosol property dataset from Polarization and Anisotropy of Reflectances for Atmospheric sci-

**Table 2.** Parameters of PSD function and refractive indices of the DT-land aerosol types according to Levy et al. (2007).

| Type | Mode | $r_v$ (μm) | $\sigma$ | $V_0$ (μm³ μm⁻²) | Refractive index |
|---|---|---|---|---|---|
| Moderately absorbing | 1 | $0.0203\tau + 0.145$ | $0.1365\tau + 0.3738$ | $0.1642\tau^{0.7747}$ | $1.43 - (-0.002\tau + 0.008)\,i$ |
| | 2 | $0.3364\tau + 3.101$ | $0.098\tau + 0.7292$ | $0.1482\tau^{0.6846}$ | $1.43 - (-0.002\tau + 0.008)\,i$ |
| Weakly absorbing | 1 | $0.0434\tau + 0.1604$ | $0.1529\tau + 0.3642$ | $0.1718\tau^{0.8213}$ | $1.42 - (-0.0015\tau + 0.007)\,i$ |
| | 2 | $0.1411\tau + 3.3252$ | $0.1638\tau + 0.7595$ | $0.0934\tau^{0.6394}$ | $1.42 - (-0.0015\tau + 0.007)\,i$ |
| Strongly absorbing | 1 | $0.0096\tau + 0.1335$ | $0.0794\tau + 0.3834$ | $0.1748\tau^{0.8914}$ | $1.51 - 0.02\,i$ |
| | 2 | $0.9489\tau + 3.4479$ | $0.0409\tau + 0.7433$ | $0.103\tau^{0.6824}$ TS1 | $1.51 - 0.02\,i$ |
| Spheroidal (dust) | 1 | $0.1416\tau^{-0.0519}$ | $0.7561\tau^{0.148}$ | $0.0871\tau^{1.026}$ | $1.48\tau^{-0.021} - 0.0025\tau^{0.132}$ TS2 $i$ (0.47) $1.48\tau^{-0.021} - 0.002\,i$ (0.55) $1.48\tau^{-0.021} - 0.0018\tau^{-0.08}$ TS3 $i$ (0.66) $1.46\tau^{-0.040} - 0.0018\tau^{-0.30}$ TS4 $i$ (2.1) |
| | 2 | 2.2 | $0.554\tau^{-0.0519}$ | $0.6786\tau^{1.0569}$ | The same as for Mode 1 |

**Table 3.** Parameters of PSD function and refractive indices of DT-ocean aerosol fine and coarse modes according to Remer et al. (2005). Details of the table below can be found at https://darktarget.gsfc.nasa.gov/algorithm/ocean/aerosol-models (last access: November 2022).

| Mode | $r_g$ | $\sigma$ | RI (0.466–0.857 μm) | RI (1.241 μm) | RI (1.628 μm) | RI (2.113 μm) |
|---|---|---|---|---|---|---|
| F1 | 0.07 | 0.4 | $1.45 - 0.0035i$ | $1.45 - 0.0035i$ | $1.45$ TS5 $-0.01i$ | $1.40 - 0.005i$ |
| F2 | 0.06 | 0.6 | $1.45 - 0.0035i$ | $1.45 - 0.0035i$ | $1.45$ TS6 $-0.01i$ | $1.40 - 0.005i$ |
| F3 | 0.08 | 0.6 | $1.40 - 0.0020i$ | $1.40 - 0.0020i$ | $1.39 - 0.005i$ | $1.36 - 0.003i$ |
| F4 | 0.10 | 0.6 | $1.40 - 0.0020i$ | $1.40 - 0.0020i$ | $1.39 - 0.005i$ | $1.36 - 0.003i$ |
| C1 | 0.40 | 0.6 | $1.35 - 0.001i$ | $1.35 - 0.001i$ | $1.35 - 0.001i$ | $1.35 - 0.001i$ |
| C2 | 0.60 | 0.6 | $1.35 - 0.001i$ | $1.35 - 0.001i$ | $1.35 - 0.001i$ | $1.35 - 0.001i$ |
| C3 | 0.80 | 0.6 | $1.35 - 0.001i$ | $1.35 - 0.001i$ | $1.35 - 0.001i$ | $1.35 - 0.001i$ |
| C4 | 0.60 | 0.6 | $1.53 - 0.003i$ (0.47) $1.53 - 0.001i$ (0.55) $1.53 - 0.000i$ (0.65) $1.53 - 0.000i$ (0.86) | $1.46 - 0.000i$ | $1.46 - 0.001i$ | $1.46 - 0.000i$ |
| C5 | 0.50 | 0.8 | The same as for Mode C4 | $1.46 - 0.000i$ | $1.46 - 0.001i$ | $1.46 - 0.000i$ |

ence coupled with Observations from a Lidar (PARASOL) (Dubovik et al., 2014).

This aerosol type was also used to calculate the top-of-atmosphere (TOA) reflectance of aerosol-contaminated clouds using the SCIATRAN software and to support the derivation of AOT above cloud over western Africa (Mei et al., 2019). Moreover, the possibility of combining the dust aerosol type with other aerosol types (e.g., WMO type) implemented in SCIATRAN was successfully demonstrated by Mei et al. (2019).

The full release of this new implementation will be included in the upcoming SCIATRAN version.

### 4.1.5 MERRA aerosol type

A mismatch between the satellite-derived and model-simulated aerosol properties exists due to different treatments of aerosol types in the satellite and model communities. In fact, aerosol type and the corresponding optical properties in the model community typically depend on time, geographic location and altitude, whereas the dependence of aerosol type on the altitude is not accounted for in the satellite community. To minimize this mismatch, we proposed a new global aerosol-type parameterization for the XBAER algorithm, which characterizes aerosol-type parameterization using the aerosol optical properties based on the MERRA aerosol components. As the first step, these components are included in the SCIATRAN software.

**Table 4.** Parameters of PSD function and refractive index of XBAER-ocean aerosol types according to Mei et al. (2018b).

| Type | Mode | $r_v$ (μm) | $\sigma$ | $V_0$ (μm³ μm⁻²) | Refractive index |
|---|---|---|---|---|---|
| Pure maritime | 1 | $0.115\tau_f^{-0.05}$ | $0.42\tau_f^{-0.02}$ | $0.078\tau$ | $1.415 - 0.002i$ |
| | 2 | $1.376\tau_f^{-0.06}$ | $0.732\tau_f^{0.016}$ | $0.588\tau + 0.002$ | $1.363 - 3 \times 10^{-9}i$ |
| Pollution 1 | 1 | $0.042\tau + 0.137$ | $0.435\tau^{0.076}$ | $0.144\tau + 0.009$ | $1.48 + 0.053\tau - (-0.002\tau + 0.006)i$ |
| (organic/black carbon) | 2 | $1.122\tau + 1.528$ | $0.658\tau_f^{-0.05}$ TS7 | $0.059\tau^{-0.18}$ | $1.48 + 0.053\tau - (-0.002\tau + 0.006)i$ |
| Pollution 2 | 1 | $0.053\tau + 0.144$ | $0.499\tau^{0.092}$ | $0.141\tau + 0.015$ | $1.419 + 0.037\tau - (-0.002$ TS8 $\tau + 0.005)i$ |
| (sulfates) | 2 | $0.31\tau + 2.195$ | $0.6\tau^{-0.06}$ | $0.115\tau + 0.018$ | $1.419 + 0.037\tau - (-0.002$ TS9 $\tau + 0.005)i$ |
| Dust-influenced | 1 | $0.048\tau + 0.117$ | $0.556\tau_f^{-0.03}$ | $0.035\tau + 0.01$ | $0.0025\tau^{0.132}$ |
| (dust) | 2 | $0.072\tau_f + 1.622$ | $0.531\tau^{-0.08}$ | $0.617\tau^{1.15}$ | $0.0025\tau^{0.132}$ |

The MERRA model contains 15 aerosol tracers (components): sulfate (SU), hydrophobic and hydrophilic modes of organic carbon (OC) and black carbon (BC) aerosol and five non-interacting size bins for dust (DU) and sea salt (SS) aerosol (see Table 5). Each aerosol component is defined by a fixed size distribution. In particular, the size distributions for SU, OC and BC components are described by a lognormal distribution function. The optical database for each tracer, $i$, includes extinction, $Q_{ei}$, and scattering, $Q_{si}$, efficiency and expansion coefficients of scattering matrix elements precalculated at predefined wavelength and humidity grids. Additionally, the database includes cross-sectional area, $A_i$, and particle mass, $m_i$, at predefined humidity grids.

The extinction cross section is calculated as

$$\sigma_{ei}(\lambda, h) = Q_{ei}(\lambda, h)\, A_i(h)\, m_i(0), \tag{6}$$

where $h$ is the humidity and $m_i(0)$ is the dry mass of tracer $i$, $i = 1, 2, \ldots, 15$. Here and thereafter, cross-sectional area, $A_i$, is given per kilogram dry mass. Replacing $Q_{ei}(\lambda, h)$ in Eq. (6) with $Q_{si}(\lambda, h)$, the scattering cross section, $\sigma_{si}(\lambda, h)$, for tracer $i$ is obtained.

In order to calculate the extinction coefficient, one needs to calculate the tracer number density:

$$N_i(z) = \frac{r_i(z)\, \rho_{air}(z)}{m_i(0)}, \tag{7}$$

where $r_i(z)$ is the mass-mixing ratio for the dry air, $\rho_{air}(z)$ is the air density and $z$ is the altitude. The air densities and mass-mixing ratios for all the tracers are obtained from the MERRA input data file. The extinction coefficient of tracer $i$ is given by

$$e_i(\lambda, z) = \sigma_{ei}(\lambda, h_z)\, N_i(z), \tag{8}$$

and the extinction coefficient of the MERRA aerosol type is

$$e(\lambda, z) = \sum_{i=1}^{15} \sigma_{ei}(\lambda, h_z)\, N_i(z), \tag{9}$$

where $\sigma_{ei}(\lambda, h_z)$ is obtained after interpolation of $\sigma_{ei}(\lambda, h)$ given by Eq. (6) to the humidity at altitude $z$, which is given in the MERRA input data file. The scattering coefficient of tracer $i$, $s_i(\lambda, z)$, and the total scattering coefficient, $s(\lambda, z)$, are calculated similarly to Eqs. (8) and (9), replacing $\sigma_{ei}(\lambda, h_z)$ by $\sigma_{si}(\lambda, h_z)$. The phase function (scattering matrix element (1,1)) for the MERRA aerosol type is calculated as follows:

$$F(\lambda, z) = \frac{1}{s(\lambda, z)} \sum_{i=1}^{15} s_i(\lambda, z)\, F_i(\lambda, z), \tag{10}$$

where $F_i(\lambda, z)$ is a scattering matrix element of tracer $i$.

In order to use the MERRA aerosol type within SCIATRAN, the user needs to provide a mass-mixing ratio (MMR) file for a given time and geographical location. To facilitate the usage of this aerosol type for inexperienced users, MMR files for typical aerosol scenarios (e.g., dust, biomass burning, urban pollution and sea salt) are included in the SCIATRAN database. Details of how to prepare the MMR file as a SCIATRAN input for a specific time and geographical location can be found in the user guide and in headers of MMR files for typical aerosol scenarios.

The MERRA aerosol type is operationally used to derive the NASA MERRA aerosol property dataset. The MERRA aerosol data products are widely used in the Intergovernmental Panel on Climate Change (IPCC) report (Randles et al., 2017).

Implementations of other aerosol types in SCIATRAN are ongoing, for instance, the use of Aerosol Robotic Network (AERONET) measurement data (e.g., RI and PSD) as direct input for the radiative transfer simulations. The users are also encouraged to send their own ideas and requests to us.

### 4.2 Clouds and snow

The SCIATRAN software package includes a water droplet and ice crystal optical parameter database (scattering matrices, scattering and extinction cross sections), which enables

**Table 5.** Maximal and minimal radii of the particle size bins for dry sea salt and dust.

| Bin number | Dry sea salt | | Dust | |
|---|---|---|---|---|
| | $r_{min}$ (µm) | $r_{max}$ (µm) | $r_{min}$ (µm) | $r_{max}$ (µm) |
| 1 | 0.03 | 0.1 | 0.1 | 1.0 |
| 2 | 0.1 | 0.5 | 1.0 | 1.8 |
| 3 | 0.5 | 1.5 | 1.8 | 3.0 |
| 4 | 1.5 | 2.0 | 3.0 | 6.0 |
| 5 | 5.0 | 10.0 | 6.0 | 10.0 |

us to calculate radiative transfer through liquid water, ice, mixed-phase clouds and snow layers. Additionally, an absorber (e.g., soot) within the cloud layer (Rozanov et al., 2014) can be accounted for by adding a specific aerosol type.

### 4.2.1 Liquid water cloud database

The version of SCIATRAN described by Rozanov et al. (2014) includes optical parameters (scattering matrices, extinction and scattering cross sections) of water droplets with effective radii between 4 and 20 µm (in step 2 µm) in the spectral range from 0.2 to 40 µm. The new implementation extends the range of the effective radius to 2–40 µm. The optical properties of liquid water clouds are precalculated using the Mie code (Mishchenko et al., 1999). The refractive index of pure water was selected according to Segelstein (1981). The water droplet size distribution was assumed to be gamma distribution:

$$f(r) = \frac{N_d r^\eta}{\Gamma(\eta+1)} \left(\frac{\eta}{r_m}\right)^{\eta+1} \exp\left(-r\frac{\eta}{r_m}\right), \qquad (11)$$

where $N_d$ is the number density of water droplets, $r_m$ and $\eta$ are the mode radius and half-width parameter, respectively, and $\Gamma(\eta+1)$ is the gamma function. As in the previous version of SCIATRAN, the half-width parameter is set to 6. Recall that the relationship between mode radius and effective radius, $r_e$, is given by $r_e = r_m(1+3/\eta)$ (Kokhanovsky, 2006). The SCIATRAN software offers the possibility of introducing a vertical inhomogeneity within a liquid water cloud considering $N_d$ and $r_e$ to be functions of altitude. At the same time, the vertical coordinate, $z$, within a cloud is defined by employing the concept of dimensionless "altitude" as suggested by Feigelson (1981); i.e., it is introduced as $x = (h_t-z)/(h_t-h_b)$, where $h_t$ and $h_b$ are the top and bottom heights of the cloud layer, respectively. Thus, to introduce an inhomogeneity, the user needs to specify $N_d(x)$ and $r_m(x)$ as functions of the dimensionless coordinate $x$, which ranges from 0 to 1.

The water droplet database within the SCIATRAN software was employed, e.g., to derive the cloud optical thickness and water droplet effective radius (Mei et al., 2019) and to retrieve the vertical profile of the droplet effective radius (Kokhanovsky and Rozanov, 2012).

### 4.2.2 Ice cloud and snow layer database

The previous ice cloud database in SCIATRAN only included optical parameters of two ice crystal shapes, i.e., fractal of second generation (Macke et al., 1996) and hexagonal prism. Moreover, the optical characteristics were precalculated within a limited range of ice crystal size. In particular, for second-generation fractal the tetrahedron side length was set to 50, 100 and 300 µm, which corresponds to the effective radii of 23, 46 and 142 µm, respectively. For hexagonal prism, the optical properties were precalculated for a hexagonal prism height of 100 µm and side lengths of 12.5, 25 and 50 µm.

The limited range of ice crystal sizes can introduce large uncertainties into the ice cloud effective radius retrieval (Mei et al., 2018a). To overcome this problem, ice crystal databases developed by Baum et al. (2011) and Yang et al. (2013) were also included in the SCIATRAN software. These ice cloud databases were operationally used to derive the NASA cloud property dataset from MODIS (King et al., 2013; Platnick et al., 2017).

The Baum ice crystal database contains optical parameters of the "AggregateSolidColumns" shape of ice crystals in the spectral range 0.2–99 µm with effective radii between 5 and 60 µm and a step of 2.5 µm.

The Yang database contains optical parameters of 11 ice crystal shapes, following Pohl et al. (2020), and 9 shapes are included in the SCIATRAN software, i.e., an aggregate of 8 columns, droxtal, hollow bullet rosettes, a hollow column, a plate, an aggregate of 5 plates, an aggregate of 10 plates, solid bullet rosettes, and a column. The optical parameters include extinction efficiency, extinction cross section, single-scattering albedo and six elements of the scattering matrix precalculated for the maximal dimension of ice habits between 2 and 10 000 µm in the spectral range from 0.2 to 15.25 µm. The extinction cross section, single-scattering albedo, nonzero elements of the scattering matrix, effective radius, projected area and volume of each habit in 189 dimensions were used from the original Yang database. The expansion coefficients were calculated by expanding the elements of the scattering matrix in the generalized spherical functions. Additional details and some applications can be found in Pohl et al. (2020).

It is worth noting that, for the same maximal dimension of ice crystal, its effective radius depends on its shape. For instance, the maximal dimension of 700 µm results in an effective radius of 295.3 µm for a droxtal, while it is 53.0 µm for a plate. The relationship between maximal dimension and effective radius for crystals of different shapes is reported by Yang et al. (2013) and also included in the SCIATRAN database.

Natural ice cloud layers are usually composed of polydisperse ice crystals of different shapes (Baum et al., 2005). For the Yang database, the polydispersity and habit mixture can be considered in SCIATRAN using a PSD function and a de-

sired habit ratio, respectively. In particular, to describe the polydisperse properties of a habit mixture, the gamma distribution with respect to the maximal dimension, $D$, following Saito et al. (2019), is used:

$$n(D) = N_c\, C\, (D/D_0)^{k-1} \mathrm{e}^{-(k-1)\, D/D_0} \ . \tag{12}$$

Here, $N_c$ is the number of ice crystals per unit volume, $k$ and $D_0$ are the shape and mode parameters, and $C$ is the normalization factor. The number density, $N_c$, and mode, $D_0$, can be selected as constant within an ice cloud layer or depending on the altitude. This enables users to formulate a realistic model of a vertically inhomogeneous, polydisperse crystal habit mixture of an ice cloud and snow layer.

Similarly to the snow grain habit mixture model used by Saito et al. (2019), the habit mixture implemented in SCIATRAN depends on the dimensions of ice crystals. In particular, a user can predefine mixing ratios $f_h(D_i)$ of any aforementioned ice particle habit, $h$, at a specific dimension grid point $D_i$. SCIATRAN interpolates the mixing ratios linearly (another interpolation type is not available) with respect to $\ln D$ between division points (as recommended in Saito et al., 2019) and calculates the bulk optical properties of an ice particle habit mixture by weighting the single-habit properties according to their mixing ratio. Following Baum et al. (2011), the scattering and extinction cross sections as well as the scattering matrix are calculated as

$$\sigma_p = \sum_{h=1}^{N_h} \int_{D_1}^{D_2} f_h(D)\, \sigma_{p,h}(D)\, n(D)\, \mathrm{d}D \ , \tag{13}$$

$$\mathbf{F}(\gamma) = \frac{1}{\sigma_s} \sum_{h=1}^{N_h} \int_{D_1}^{D_2} f_h(D)\, \mathbf{F}_h(\gamma, D)\, \sigma_{s,h}(D)\, n(D)\, \mathrm{d}D \ , \tag{14}$$

where $N_h$ is the number of habits, the subscript $p$ denotes "s" or "e" for the scattering, $\sigma_{s,h}(D)$, or extinction, $\sigma_{e,h}(D)$, cross sections, respectively, $\mathbf{F}_h(\gamma, D)$ is the scattering matrix of habit $h$ with dimension $D$, and

$$\sum_{h=1}^{N_h} f_h(D) = 1 \ . \tag{15}$$

The integrals in Eqs. (13) and (14) are calculated by SCIATRAN numerically using the trapezoidal rule, and partitioning of the interval $[D_1 = 2\,\mu\mathrm{m}, D_2 = 10\,000\,\mu\mathrm{m}]$ is performed according to the discrete grid used by Yang et al. (2013). In particular, the intervals are 1, 2.5, 12.5 and 125 $\mu\mathrm{m}$ for the ranges [2–10], [10–100], [100–1000] and [1000–10 000] $\mu\mathrm{m}$, respectively.

As mentioned above, the SCIATRAN software enables the user to introduce a vertical inhomogeneity within an ice cloud or snow layer considering $N_c$ and $D_0$ to be functions of depth. In this case the approach of dimensionless "altitude" is employed as described above for a liquid water cloud. Thus,

the user needs to specify $N_c(z)$ and $D_0(z)$ as functions of the dimensionless coordinate $x$, which ranges from 0 to 1.

For a vertically homogeneous monodisperse habit mixture cloud, the input parameters include the optical thickness of the cloud, maximal dimension and fractions of the selected ice crystal shapes. Combining the liquid water and ice crystal optical properties, the radiative transfer calculations can also be performed for a mixed-phase cloud.

The database of the optical properties of ice crystals proposed by Yang et al. (2013) was successfully utilized to (1) estimate the impact of habit mixture on the simulated TOA reflectance from the POLarization and Directionality of the Earth's Reflectances (POLDER) instrument (Pohl et al., 2020) and to (2) derive snow property research products (snow grain size and shape) using TOA reflectance measurements from the Copernicus Sentinel-3 Sea and Land Surface Temperature Radiometer (SLSTR) instrument in the framework of the XBAER algorithm (Mei et al., 2021a, b).

Although it has been demonstrated that the database of the optical properties of ice crystals proposed by Yang et al. (2013) can be used for snow studies, a realistic model of a snow layer is usually represented by a vertically inhomogeneous, polydisperse ice crystal habit mixture (Mei et al., 2021a), which requires a priori knowledge of several properties of the snow layer (e.g., appropriate numbers of habits and their fraction as well as parameters of the PSD function).

To simplify the requirements for such a priori knowledge, we have also implemented in the SCIATRAN software a stochastic mixture model of ice crystals and air gaps based on the stereological approach (Malinka, 2014). In the framework of the stereological approach (geometrical statistic), the snow layer is considered to be a two-phase random mixture of ice particles and air gaps. The grains of irregular shapes are characterized in that case by random chord length distribution. Under a random chord we understand a straight line connecting any two points on the border inside the ice crystal or air gap. The chord length is a random value which is determined by some distribution function. With an assumption of a stochastic ice crystal mixture, this approach uses the concept of the chord length distribution rather than the characteristics of a separate crystal. The theoretical background to calculate the optical parameters of the corresponding stochastic snow model can be found in Malinka (2014, 2015).

The usage of a stochastic model has at least two advantages as compared to the Yang et al. (2013) database. First, the stochastic model does not require any database, because the optical parameters can be easily calculated online. Second, for a vertically homogeneous snow layer, users only need to have information about the mean chord of the ice crystals and the optical thickness of the snow. The mean chord length plays the role of the effective size of an ice crystal; i.e., the mean chord coincides with the factor of 0.75 with the standard definition of the effective radius (Malinka, 2014).

**Table 6.** Mean chords and optical thicknesses of different surface types (Malinka et al., 2016).

| Surface type | Mean chord (μm) | Optical thickness |
|---|---|---|
| Bright white ice | 450–720 | 14–50 |
| Typical white ice | 2200–2800 | 9–20 |
| Melting ice | 4700–10 000 | 2.2–5.4 |
| Snow-covered ice | 170–270 | 27–73 |

We recall, however, that the usage of a stochastic model is reasonable if the sizes of ice particles are significantly larger than the wavelength. Retrieved values of mean chords and optical thicknesses of different white ice types according to Malinka et al. (2016) are summarized in Table 6.

## 4.3 Surface reflection

Surface albedo and BRDF are important parameters in RT calculations. The SCIATRAN software includes both Lambertian albedo and various BRDF models (Rozanov et al., 2014). In the case of Lambertian reflection, the user can select a spectrally dependent or constant albedo. In particular, a global dataset of vegetation and land albedo (Matthews, 1983) is available in the SCIATRAN software. For RT calculations accounting for a surface BRDF, users can choose one of the following built-in models: RPV (Rahman et al., 1993), kernel-based Ross–Li (also called RTLSR) (Maignan et al., 2004; Roujean et al., 1992), kernel-based Ross–Li vegetation–soil (Breon and Vermote, 2012) and modified RPV plus facet (Diner et al., 2012). A brief description of the aforementioned models is given in Appendix B.

For a retrieval of surface properties from satellite observations, the following surface reflectance models are additionally included in the SCIATRAN software: PROSAIL, XBAER, snow and melt ponds on sea ice.

In order to facilitate users in selecting a proper BRDF model, the following general recommendations can be formulated.

- The multi-parameter PROSAIL model is recommended to investigate the sensitivity of surface and TOA reflectances with respect to the biophysical vegetation parameters on a local or regional scale.

- The single-parameter XBAER model can be advised for investigations of surface and TOA reflectances on a global scale because only one parameter is needed for this model.

- The RPV and RTLSR models are recommended for modeling of multi-angle scalar observations of BRDF in case of a snow-free land surface.

- The modified RPV plus facet model is suitable for modeling multi-angle polarized observations of BRDF in case of a snow-free land surface.

- The FASMAR and RTLSRS models are recommended for simulations of multi-angle scalar observations of BRDF for snow-covered surfaces.

### 4.3.1 PROSAIL model for vegetated surfaces

The PROSAIL model is a combination of the PROSPECT and SAIL (Scattering by Arbitrary Inclined Leaves) models (Jacquemound et al., 2009). PROSAIL is used to describe the directional reflectance of the plant canopy and to retrieve biophysical properties of vegetation.

The PROSAIL source code (http://teledetection.ipgp.jussieu.fr/prosail/, last access: November 2022) was adopted and implemented in the SCIATRAN software. The input parameters are leaf inclination distribution function type, leaf area index (LAI), hotspot parameter, chlorophyll content, carotenoid content, brown pigment content, equivalent water thickness (EWT), leaf mass per area (LMA), structure coefficient and soil factor.

Using a desired model of the atmosphere in combination with BRDF provided by the implemented PROSAIL model, the user can calculate the radiation field in the atmosphere in the UV, VIS and NIR spectral ranges, accounting for the angular scattering properties of a vegetated surface. To verify the implementation of the PROSAIL model, a comparison of TOA reflectances simulated by SCIATRAN and measured by the MODIS instrument is presented in Sect. 6 (see Fig. 4a).

### 4.3.2 XBAER land surface reflectance parameterization

The XBAER model describes the surface reflectance by a linear relationship with respect to the Soil-Adjusted Vegetation Index (SAVI) (Mei et al., 2017a). This model was originally designed for AOT retrieval in the framework of the XBAER algorithm (Mei et al., 2017a) using measurements of the MERIS instrument. For wavelengths above the red edge (MERIS channels 10, 12, 13 and 14, which correspond to 753, 778, 865 and 885 nm), the surface reflectance, $R(\lambda_i)$, was approximated as

$$R(\lambda_i) = a(\lambda_i)\, x + b(\lambda_i), \quad i = 10, 12, 13, 14, \tag{16}$$

where $i$ is the MERIS channel number, $a(\lambda_i)$ and $b(\lambda_i)$ are the slope and intercept for given wavelengths $\lambda_i$ in the Surface Reflectance Wavelength Shape Constrain Database (SRWSCD) (Mei et al., 2017a) and $x$ is the SAVI.

For wavelengths below the red edge (MERIS channels 1, 2,…, 9, which correspond to 412, 443, 490, 510, 560, 620, 665, 681 and 708 nm), the ratio between the surface reflectance at each wavelength, $\lambda_i$, and the surface reflectance at MERIS channel 14 was parameterized by

$$R(\lambda_i)/R(\lambda_{14}) = \tilde{a}(\lambda_i)\, x + \tilde{b}(\lambda_i), \quad i = 1, 2, \ldots, 9. \tag{17}$$

The SRWSCD database, which belongs to the XBAER retrieval algorithm and is not included in the SCIATRAN

database, includes the geographically dependent monthly slope and intercept of the linear model.

Regression coefficients for typical surface types (e.g., vegetated surface, urban, desert) are included in the SCIATRAN database. Using the data files for these typical surface types as templates, users can prepare their own regression coefficients following Mei et al. (2017a).

A comparison of simulated TOA reflectances obtained utilizing XBAER parameterization and the PROSAIL model is presented in Sect. 6 (see Fig. 4a).

### 4.3.3   Snow and melt ponds on sea ice

Besides the modified RPV (Degunther and Meerkotter, 2000) and the asymptotic model (Kokhanovsky and Zege, 2004) of the snow reflectance, the recent snow-kernel-based BRDF models, such as RTLSRS and FASMAR suggested by Jiao et al. (2019) and Mei et al. (2022), respectively, as well as the model of white ice and melt ponds on sea ice (Malinka et al., 2016, 2018), were also included.

#### RTLSRS model

The kernel-driven RTLSRS BRDF model with a snow kernel (Jiao et al., 2019) was implemented in the following form:

$$\rho(\Omega, \lambda) = f_i(\lambda) + f_v(\lambda) K_v(\Omega) + f_g(\lambda) K_g(\Omega) + f_s(\lambda) K_s(\Omega) , \quad (18)$$

where $f_i$, $f_v$, $f_g$ and $f_s$ are wavelength-dependent coefficients, $\Omega$ comprises angular variables $\{\vartheta_0, \vartheta, \varphi\}$, which describe the solar zenith angle, viewing zenith angle and relative azimuth, respectively, $\lambda$ is wavelength, and $K_v$, $K_g$ and $K_s$ are the volumetric scattering kernel, geometric–optical kernel and snow kernel, respectively.

As compared to the classical Ross–Li model suggested by Lucht et al. (2000), an additional snow kernel based on the asymptotic model of Kokhanovsky and Zege (2004) was introduced to describe the strong forward-scattering peak typical of the snow reflection. Although the SCIATRAN software does not contain a database of the coefficients $f_i(\lambda)$, $f_v(\lambda)$, $f_g(\lambda)$ and $f_s(\lambda)$, the kernels are included. The user needs to provide a file containing the coefficients at a discrete number of wavelengths. Some examples of the coefficients obtained by fitting the RTLSRS model to the measured TOA reflectance are presented in Jiao et al. (2019). The file structure is described in the user guide.

#### FASMAR

In contrast to the RTLSRS model, where the snow kernel based on the Asymptotic Radiative Transfer (ART) model (Kokhanovsky and Zege, 2004) was used to improve the accuracy in the case of a snow reflection, FASMAR exploits the ART model as a main kernel. In particular, FASMAR (see Mei et al., 2022, for details) includes four kernels, and

the model is formulated as

$$\rho(\Omega, \lambda) = f_1 + f_2 K_2(\Omega, \lambda) + f_3 K_3(\Omega, \lambda) + f_4 K_4(\Omega) + f_5 K_5(\Omega) , \quad (19)$$

where $f_1, f_2, \ldots, f_5$ are wavelength-dependent parameters (argument $\lambda$ is omitted), and the kernels are given by

$$K_2(\Omega, \lambda) = R_\infty^0(\Omega) \, e^{-\beta_\lambda A(\Omega)} - k_2(\lambda) , \quad (20)$$

$$K_3(\Omega, \lambda) = p_\lambda(\theta)/(\mu + \mu_0)/4 - k_3(\lambda) , \quad (21)$$

$$K_4(\Omega) = R_\infty^0(\Omega) \, e^{\cos\theta} - k_4 , \quad (22)$$

$$K_5(\Omega) = \cos\theta - k_5 . \quad (23)$$

On the right-hand side of each equation the second term is introduced which modifies the respective kernel to obtain $K_2 = K_3 = K_4 = K_5 = 0$ for SZA = 0° and VZA = 0°. The modification terms are given by

$$k_2(\lambda) = R_\infty^0(\tilde\Omega) \, e^{-\beta_\lambda A(\tilde\Omega)} , k_3(\lambda) = \frac{p_\lambda(-1)}{8} ,$$

$$k_4 = \frac{R_\infty^0(\tilde\Omega)}{2.718} , k_5 = -1 . \quad (24)$$

As a result, the coefficient $f_1$ gives the magnitude of the reflectance in the nadir direction (Roujean et al., 1992). In Eqs. (20)–(24), $\mu = \cos\vartheta$, $\mu_0 = \cos\vartheta_0$, $\vartheta_0$, $\vartheta$ and $\theta$ are the solar zenith angle, viewing zenith angle and scattering angle, respectively, $\tilde\Omega = \{0, 0, 0\}$, $\beta_\lambda = \sqrt{\gamma_\lambda d}$, $\gamma_\lambda = 4\pi \kappa_i/\lambda$ is the absorption coefficient of ice, $\kappa_i$ is the imaginary part of the ice refractive index, $d$ is an effective grain size, defined by the ratio of the average volume to the average surface area of grains, $p_\lambda(\theta)$ is the phase function, and $R_\infty^0(\Omega)$ is the reflection function of semi-infinite non-absorbing media.

FASMAR includes the ART kernel, $K_2$, a single-scattering kernel, $K_3$, and auxiliary kernels, $K_4$ and $K_5$. The kernel $K_5$ is only used for wavelengths longer than 1600 nm. In kernel $K_2$, the function $R_\infty^0(\Omega)$ is implemented according to the approximation suggested by Kokhanovsky (2005), and the function $A(\Omega)$ is given by $A(\Omega) = b_n K_0(\mu) K_0(\mu_0)/R_\infty^0(\Omega)$ (Kokhanovsky and Zege, 2004), where $K_0(\mu)$ is called the escape function, which is implemented according to the common approximation (Sobolev, 1972) as $K_0(\mu) = 3/7(1 + 2\mu)$. The parameter $b_n$ depends mainly on the ice crystal shape. For the fractal ice particles, it was estimated by Kokhanovsky and Zege (2004) to be 3.62. As a result, the final expression for $A(\Omega)$ is given by $A(\Omega) = 0.66(1 + 2\mu)(1 + 2\mu_0)/R_\infty^0(\Omega)$.

Similarly to the RTLSRS model, the SCIATRAN software does not contain a database of the coefficients $f_1, f_2, \ldots, f_5$. The user needs to provide a file containing the coefficients at a discrete number of wavelengths. The file structure is described in the user guide.

Comprehensive comparisons of the FASMAR and RTLSRS models with ground-based, aircraft and satellite measurements of snow angular reflectance can be found in Mei

et al. (2022). FASMAR can be used for large solar zenith angles (e.g., SZA > 80°, which is a typical case for the polar regions), while both the Ross–Li and RTLSRS models are recommended for use for SZAs < 70°.

## 5   White ice and snow-covered ice reflectance model

White ice and snow-covered ice are two typical surface types in the polar regions. Following Malinka et al. (2016), the white ice BRDF model describes the reflection of an ice layer with a highly scattering granular layer on top. Since the grain size of ice crystals is significantly larger than the wavelength of incident light, the optical parameters (extinction coefficient, SSA and phase function) of a stochastic mixture of irregular ice crystals and air gaps were analytically derived by Malinka (2014, 2015). Using the analytical expressions for optical parameters and employing asymptotic radiative transfer theory, the following analytical expression for the BRDF of white ice was proposed by Malinka et al. (2016) and implemented in SCIATRAN:

$$R = R_\infty^0 \, \frac{\sinh\left\{\gamma\,\tau + y\,[1 - K(\vartheta)\,K(\vartheta_0)/R_\infty^0]\right\}}{\sinh(\gamma\,\tau + y)} \,, \qquad (25)$$

where $R_\infty^0$ is the BRDF of a non-absorbing semi-infinite layer with the same scattering phase function as the snow layer, sinh denotes the sinus hyperbolicus, $\tau$ is the optical thickness of the snow layer and $\vartheta_0$ and $\vartheta$ are the incidence and observation polar angles, respectively:

$$\gamma = \sqrt{3\,(1-\omega)\,(1-\omega g)} \,, \; K(\vartheta) = \frac{3}{7}\,(1 + 2\cos\vartheta) \,, \quad (26)$$

$$y = 4\,\sqrt{\frac{(1-\omega)}{3\,(1-\omega g)}} \,, \; K(\vartheta_0) = \frac{3}{7}\,(1 + 2\cos\vartheta_0) \,. \qquad (27)$$

The expression for the single-scattering albedo is

$$\omega = 1 - \frac{\alpha\,n^2\,a\,T_{\mathrm{dif}}}{\alpha\,n^2\,a + T_{\mathrm{dif}}} \,, \; \alpha = \frac{4\,\pi\,\kappa}{\lambda} + \alpha_y \,, \qquad (28)$$

where $n$ and $\kappa$ are the real and imaginary parts of the ice refractive index, $a$ is the mean chord of ice crystals and $\alpha_y$ is the absorption by the yellow substance in the white ice. Analytical expressions for the asymmetry factor ($g$), $\alpha_y$, and auxiliary function $T_{\mathrm{dif}}$ are given in Appendix C.

The usage of Eq. (25) to calculate BRDF in the framework of the SCIATRAN software requires the following parameters to be provided by the user: optical thickness of the snow layer, $\tau$, mean chord of the ice crystals, $a$, and the yellow substance absorption coefficient, $\alpha_y$. The refractive index of ice can be either taken from the SCIATRAN database or provided by the user. The calculation of $R_\infty^0$ is performed by SCIATRAN using a precalculated lookup table.

## Melt ponds on the sea ice reflectance model

The BRDF model of melt ponds on sea ice (Malinka et al., 2018) was implemented in SCIATRAN as follows:

$$\begin{aligned}R_\lambda(\Omega) = \; &\frac{1}{\mu_0}\,R_{\mathrm{F}}(\mu_0)\,\delta(\mu - \mu_0)\,\delta(\varphi) \\ &+ \frac{T_{\mathrm{F}}(\mu)\,T_{\mathrm{F}}(\mu_0)\,A_{\mathrm{b}}}{\pi\,n^2[1 - A_{\mathrm{b}}\,f_{\mathrm{in}}(\tau_{\mathrm{p}})]}\,e^{-\tau_{\mathrm{p}}/\mu_0' - \tau_{\mathrm{p}}/\mu'} \,.\end{aligned} \qquad (29)$$

Here, $R_{\mathrm{F}}(\mu_0)$ and $T_{\mathrm{F}}(\mu_0)$ are the Fresnel reflectance and transmittance of a flat water surface for the cosine of the incidence angle $\mu_0$, $n$ is the refractive index of water (assuming the refractive index of air to be 1), $\mu_0'$ is the cosine of the refractive angle ($\sqrt{1 - \mu_0^2} = n\,\sqrt{1 - \mu_0'^2}$), $\tau_{\mathrm{p}} = z\,\sigma_{\mathrm{p}}$ is the optical thickness of the pond, $z$ is the pond depth and $\sigma_{\mathrm{p}}$ is the extinction coefficient of the water, equal to the sum of the water absorption ($\alpha_{\mathrm{w}}$) and the scattering ($\sigma_{\mathrm{w}}$) coefficients, and $\delta(\mu - \mu_0)$ and $\delta(\varphi)$ are the Dirac delta functions (Korn and Korn, 1968). $A_{\mathrm{b}}$ is the pond bottom albedo. The auxiliary function $f_{\mathrm{in}}(\tau_{\mathrm{p}})$ (Malinka et al., 2018) is calculated numerically by the SCIATRAN program. For simplicity reasons, the dependence of $R_{\mathrm{F}}$, $T_{\mathrm{F}}$, $A_{\mathrm{b}}$, $n$ and $\tau_{\mathrm{p}}$ on the wavelength, $\lambda$, is not indicated. The analytical expression for the albedo of an under-pond ice layer, $A_{\mathrm{b}}$, as a function of the geometric thickness of the under-pond ice, $H$, and the transport-scattering coefficient, $\sigma_{\mathrm{t}}$, is given in Appendix D.

To calculate BRDF using Eq. (29) in the framework of the SCIATRAN software, the following input parameters need to be provided by the user: geometric thickness of the under-pond ice, $H$, water layer (pond) depth, $z$, and transport-scattering coefficient, $\sigma_{\mathrm{t}}$. Following Malinka et al. (2018), the reasonable range for $\sigma_{\mathrm{t}}$ is [0.2–10] m$^{-1}$. In particular, for light and dark melt ponds, typical values for $\sigma_{\mathrm{t}}$ are $\sim 4$ and $\sim 2$ m$^{-1}$, respectively. Water absorption and scattering coefficients as well as refractive indices of water and ice can be either taken from the SCIATRAN databases or provided by the user.

## 5   Inherent optical properties of aerosol and clouds

### 5.1   Aerosol

Figure 2 shows an example of the optical properties of the aerosol types newly implemented in SCIATRAN. All the optical properties are calculated using an AOT of 0.5 at a wavelength of 0.55 µm. For MODIS-DT, a moderately absorbing aerosol type is selected. For MODIS-OC, two basic components F1 and C2 (Table 3) are selected, and the fine-mode fraction is set to 0.5. For XBAER-OC, a purely maritime aerosol type is selected. For the Dubovik dust model, the aspect ratio is set to 1. For the OPAC 4.0 dust model, the total number of mineral particles is set to 300, representing the "background desert conditions" (Koepke et al., 2015). In

the background desert conditions, the fractions of the mineral nucleation model, mineral accumulation mode and mineral coarse model are 89.79 %, 10.16 % and 0.05 %, respectively. In this paper, the same components and fractions were used for the OPAC 3.0 dust model. The MERRA aerosol type is selected for the AERONET site Rio Branco ($-67.869°$ E, $-9.957°$ N) on 16 August 2020. With the above settings, the MODIS-DT and MERRA (picked up at the same time and location), MODIS-OC and XBAER-OC (both represent the maritime aerosol type), Dubovik dust, and OPAC 3.0 and OPAC 4.0 dust (both represent the dust aerosol type) are subgroups within which optical properties are expected to be comparable.

In Fig. 2a, the wavelength-dependent AOTs calculated using different aerosol types show the same value at 550 nm because all the optical properties are scaled to $AOT = 0.5$ at this wavelength. The Ångström coefficients (in the spectral range 440–870 nm) are 1.77 and 1.81 for MODIS-DT and MERRA, $-0.19$ and $-0.11$ for MODIS-OC and XBAER-OC, and 0.56, $-0.009$ and 0.01 for Dubovik dust and OPAC 3.0 and OPAC 4.0 dust, respectively. The lines of the MODIS-DT and MERRA aerosol types almost overlap, indicating the similarity of aerosol properties. MODIS-OC and XBAER-OC show a similar spectral pattern, with AOT slightly increasing with the wavelength. This can be explained by the significant contribution of coarse-mode particles. The Dubovik dust shows a much stronger spectral gradient as compared to OPAC 4.0 dust.

Figure 2b shows the phase functions at 550 nm for the selected aerosol types. In the case of MERRA, the phase function is selected for an altitude near the surface. As expected, MODIS-DT and MERRA, MODIS-OC and XBAER-OC, Dubovik, and OPAC 3.0 and OPAC 4.0 dust have similar phase functions, respectively. Coarse-mode-dominated aerosol types (MODIS-OC, XBAER-OC, Dubovik and OPAC 4.0) show stronger forward peak scattering as compared to fine-mode-dominated aerosol types (MODIS-DT and MERRA). For the selected scenario, MODIS-DT shows a stronger forward scattering and a weaker backward scattering as compared to MERRA. MODIS-OC and XBAER-OC, with completely different parameterization strategies, show very similar phase functions. OPAC 3.0 and OPAC 4.0 show similar behavior, while the Dubovik and OPAC 4.0 dust models show relatively large difference in both forward and backward scattering as compared to other aerosol types (e.g., MODIS-DT and MERRA, XBAER-OC and MODIS-OC).

Figure 2c shows the wavelength-dependent SSA for the selected aerosol types. MODIS-DT and MERRA show large differences in the absolute SSA values; however, both demonstrate the increase in absorption with the wavelength. MODIS-OC and XBAER-OC, especially XBAER-OC, demonstrate weak absorption. The Dubovik and OPAC 4.0 dust models again show differences in both magnitude and wavelength dependence.

Figure 2d shows the wavelength-dependent asymmetry factor, $g$, for the selected aerosol types, which is a fundamental property of aerosol particles that affects the aerosol direct radiative forcing. MODIS-OC and XBAER-OC show the largest $g$ values, indicating a stronger forward scattering, while MODIS-DT and MERRA show the smallest $g$, indicating a weaker forward scattering, which is consistent with the results presented in Fig. 2b.

## 5.2 Cloud

A comprehensive consideration of the extinction efficiency, single-scattering albedo and asymmetry factor for different wavelengths, maximum dimensions and selected crystal shapes can be found in Yang et al. (2013). In the scope of this paper we consider only the phase functions of ice crystals in the visible spectral range, where, owing to weak absorption by ice and water, the scattering processes play a very important role.

Figure 3 shows the comparison of water droplet and ice crystal phase functions with campaign measurements provided by Järvinen et al. (2018). The phase functions are presented at a wavelength of 532 nm and normalized to the Arctic CLoud Observations Using airborne measurements during polar Day (ACLOUD) measurement at a scattering angle of 18° (Järvinen et al., 2018). The effective radii of water droplets and ice crystals are selected to be the global mean values of about 13.5 and 25 μm, respectively (King et al., 2013).

As expected, the phase functions of water droplets are quite different from those of ice crystals, especially in backward-scattering directions. In particular, the famous rainbow feature of the scattering by water droplets can be observed at about 138°. The large differences in the phase functions observed by different campaigns indicate the large variability of cloud optics. Large differences are observed mainly in the backward-scattering direction. For ice crystals, the aggregate of eight columns and droxtal have the strongest backward scattering, while the plate shows the weakest one. The large variability of the campaign measurements can be well-described by a mixture of different ice crystal shapes.

## 6 Simulated TOA reflectance for different CAS scenarios

In this section we consider selected applications of aerosol and cloud databases and BRDF models for calculations of TOA reflectance for a surface–atmosphere system using the SCIATRAN software. The following surface scenarios are considered below: snow, white ice and melt ponds typical of high-latitude regions and vegetated surfaces typical of low- and middle-latitude regions.

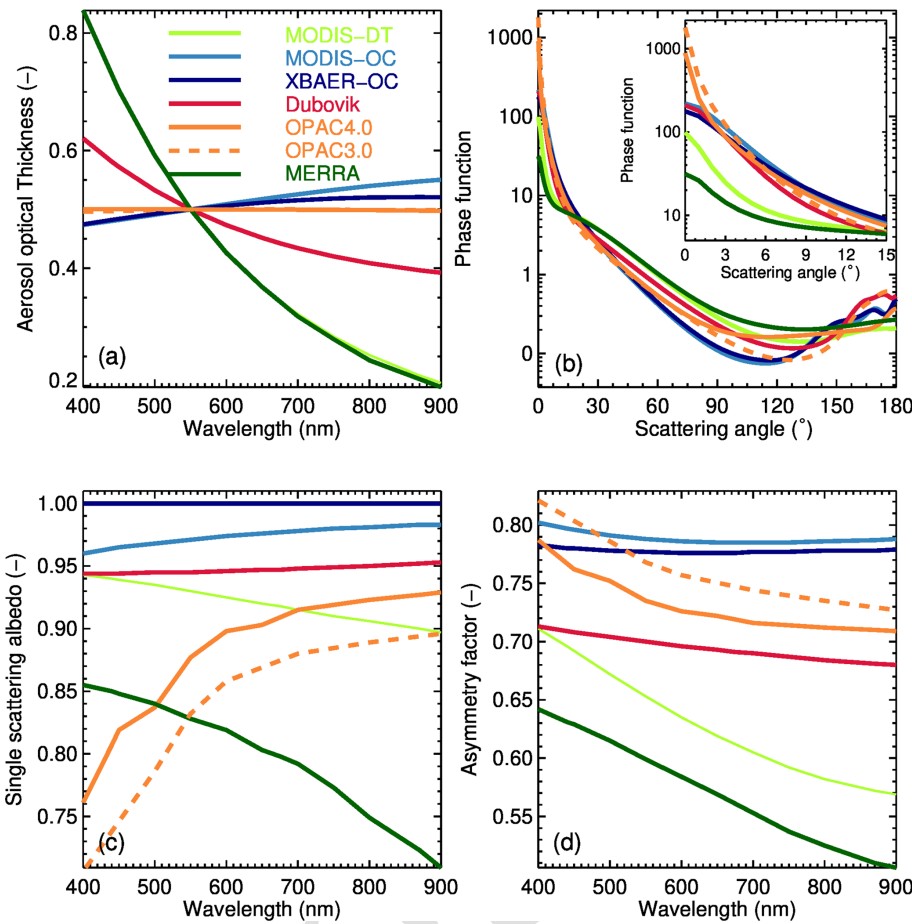

**Figure 2.** Optical properties of the aerosol types newly implemented in SCIATRAN (indicated by different colors): **(a)** wavelength-dependent AOT; **(b)** phase function at a wavelength of 0.55 μm (with a zoom-in panel in the upper-right corner); **(c)** wavelength-dependent single-scattering albedo; **(d)** wavelength-dependent asymmetry factor.

## 6.1 Vegetated surface

Simulations presented in this subsection are performed using the viewing geometry and the spectral response function of the HyperSpectral Imager (HSI) instrument on board the Environmental Mapping and Analysis Program (EnMAP) mission. HSI/EnMAP is a German perspective mission to monitor environment-related parameters launched in April 2022, and XBAER is a candidate algorithm to provide CAS products for this instrument.

Example simulations of the TOA reflectance with SCIATRAN for different scenarios were performed for the AERONET site Rio Branco ($-67.869°$, $-9.957°$) in the Amazon region for 16 August 2020. The surface reflectance was calculated by using either the XBAER parameterization or the PROSAIL model. The SAVI parameter required by the XBAER parameterization was estimated from measurements of the Ocean and Land Colour Instrument (OLCI) as 0.34. Due to database limitations, the simulations using XBAER parameterization were done only for OLCI wavelengths. The PROSAIL model was used with a LAI of 4.5, which was obtained from the MODIS LAI product. The leaf inclination distribution function type was set to Type = 1 (spherical). Other parameters are set as follows: hotspot parameter (0.01); chlorophyll content ($1 \mu g\, cm^{-2}$); carotenoid content ($8 \mu g\, cm^{-2}$); brown pigment content (0); EWT (0.01 cm); LMA ($0.009\, g\, cm^{-2}$); structure coefficient (1.5); soil factor (1.0).

The aerosol was accounted for by using the MERRA aerosol type with the fraction of each component provided by the MERRA database for the selected time and location. In particular, the AOTs of the total column, black carbon, organic carbon, sulfate, sea salt and dust are 0.241, 0.043, 0.143, 0.043, 0.003 and 0.009, respectively. A water cloud between 2.0 and 2.5 km with a cloud optical thickness (COT) of 3 and an effective radius of water droplets of 13.5 μm was used for the simulations. Along with aerosol, cloud and Rayleigh scattering, the absorption by all the relevant atmospheric gases was accounted for. The vertical profiles of pressure, temperature and absorber number density were taken

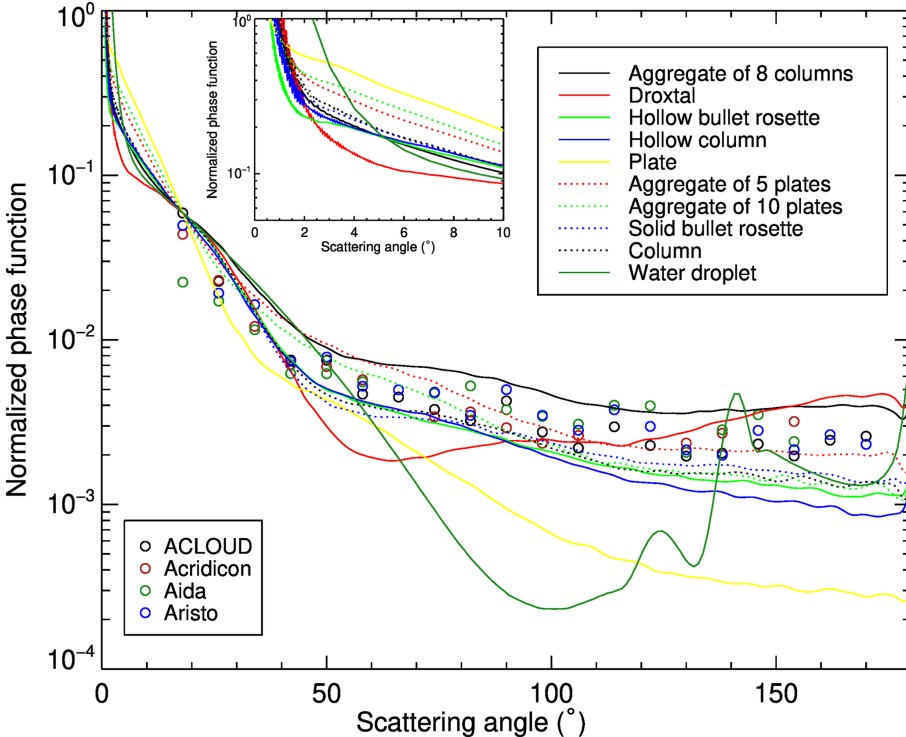

**Figure 3.** Comparison of phase functions for water droplets and ice crystals with campaign measurements provided by Järvinen et al. (2018). Symbols – measurements, lines – different shapes (see legend). A zoom-in panel for smaller scattering angles is shown in the upper-left corner of the plot.

from the database created with the Bremen 2D Chemical Transport Model (Sinnhuber et al., 2009) for August at a latitude of 10° S.

TOA reflectances simulated with SCIATRAN for different scenarios are shown in Fig. 4a. For comparison, TOA reflectances measured by the MODIS instrument at the same time and location are also presented. One can see that for clear-sky scenarios (S + R + A and S + R + A + G), the TOA reflectances simulated using both XBAER surface parameterization and the PROSAIL model agree well with reflectances observed by MODIS. The red-edge effect and sharp spectral structures are from vegetation and gas absorption, respectively. The plot indicates that employing the newly implemented features in the SCIATRAN model, the wavelength dependence and the magnitude of TOA reflectance observed by MODIS can be well-reproduced by the simulations.

For the cloudy scenario (S + R + A + G + C), one observes a general enhancement of the TOA reflectance caused by the additional reflectance of the cloud layer. However, the contribution of the surface is still visible (red edge from vegetation) because the cloud is optically thin (COT = 3).

The MERRA aerosol model used for the simulations enables us to investigate contributions of different aerosol components to the TOA reflectance. For this purpose, the TOA reflectance was calculated first, accounting for the contributions from all aerosol components (total) and then for each particular aerosol component independently. The calculated reflectances are shown in Fig. 4b. In the case under consideration (over the Amazon forest), contributions of aerosol components differ for different wavelengths, with the strongest contribution from the organic carbon component below about 800 nm.

## 6.2 Snow surface

This subsection presents example results of simulations of snow reflectance performed with the radiative transfer package SCIATRAN. The snow layer was defined as a layer with an optical thickness of 5000 and a geometrical thickness of 1 m covering a black surface. The snow layer was assumed to be vertically and horizontally homogeneous without any surface roughness and composed of monodisperse ice crystals. The impact of snow impurities and scattering/absorption processes on the atmosphere was neglected. The optical properties for the droxtal ice crystal shape with an effective radius of 126 µm (Mei et al., 2021b) from the Yang et al. (2013) database were used for the simulations.

The solar zenith angle was set to 68.9°, which corresponds to the mean value during summer at Svalbard (Mei et al., 2022).

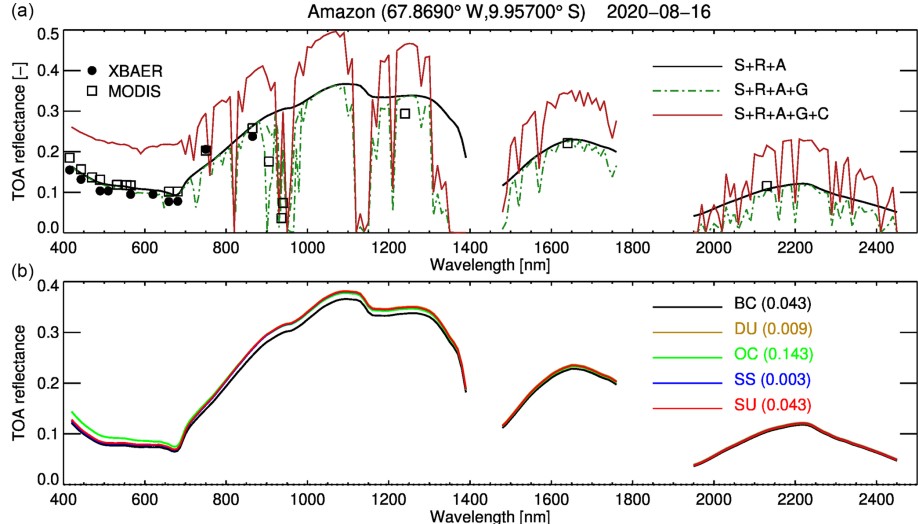

**Figure 4.** TOA reflectance simulation for a measurement from the EnMAP instrument over the Amazon region on 16 August 2020. **(a)** TOA reflectance simulated using different settings: S – surface reflectance described by the PROSAIL model, R – Rayleigh, A – aerosol, G – gaseous absorbers, C – cloud. Open rectangles depict MODIS observation data, and filled circles represent simulations with XBAER for OLCI wavelengths. **(b)** TOA reflectance simulated considering different MERRA aerosol components: BC – black carbon, DU – dust, OC – organic carbon, SS – sea salt, SU – sulfate. The AOT for each component is indicated in the parentheses.

The BRDF of the snow layer calculated with SCIATRAN is shown in the form of polar plots in the left column of Fig. 5 for wavelengths of 380, 480, 670, 870, 1220 and 2200 nm. These wavelengths are selected to represent typical spectral channels of satellite instruments (e.g., MODIS, MERIS/OLCI, AATSR/SLSTR) from UV to SWIR. For a comparison, the snow BRDF was calculated using the FASMAR and RTLSRS models. The coefficients required by these models were obtained by fitting their results to SCIATRAN simulations. The relative differences of the results from the FASMAR and RTLSRS models to those from SCIATRAN are presented in the middle and right columns of Fig. 5. One can see that reflectances obtained with the FASMAR and RTLSRS models show very good agreement (both spectral and angular) with the SCIATRAN simulations. However, the RTLSRS model shows lower reflectances in the forward direction (RAA = 0) as compared to the results from FASMAR and SCIATRAN. The relative difference between the FASMAR and SCIATRAN results is within 5 % for typical satellite observation angles. A decrease in BRDF values with increasing wavelength is seen in all three datasets. This feature is caused by the ice crystal absorption.

Further examples of FASMAR and RTLSRS model comparisons with ground-based, aircraft and satellite measurements of snow angular reflectance can be found in Mei et al. (2022).

### 6.3 White ice and snow-covered ice

Sea ice is one of the most important surface types in the high-latitude ocean regions. Measurements of spectral albedo for typical sea ice conditions were performed in the spectral range 350–1350 nm during the R/V *Polarstern* cruise ARK-XXVII/3 from 2 August to 8 October 2012 (Malinka et al., 2016). Two examples of bright white ice spectral albedo measurements, as presented in Malinka et al. (2016) (see their Fig. 8a and c), are used in this paper as well. The scattering layer for these two examples is about 3–8 cm thick, while the near-surface air temperature is about $-1.2\,°C$. Such sea ice scenarios represent an intermediate phase between typical snow and white ice (Malinka et al., 2016).

Simulations with SCIATRAN were performed using the stochastic model of the surface reflection described in Sect. 4.3.3 and an aerosol-free atmosphere without gaseous absorption. Figure 6a and b show comparisons of measured and modeled bright white ice spectral albedo with and without yellow substance absorption, respectively.

Three parameters required to initialize the BRDF model in SCIATRAN (snow layer optical thickness, mean chord and yellow substance absorption) were obtained by fitting the simulated albedo with respect to the measured one within the spectral range 350–1350 nm. The following parameters provide the minimal residual between the measured and simulated spectral albedos presented in Fig. 6a and b: ($\tau = 350.0$, $a = 510.6\,\mu m$, $\alpha_y = 0.71\,m^{-1}$) and ($\tau = 14.1$, $a = 776.2\,\mu m$, $\alpha_y = 0.001\,m^{-1}$), respectively. The spectral behavior in the visible spectral range determined by the yellow substance absorption is well-reproduced by SCIATRAN. The RMSE values between SCIATRAN simulations and measurements are $1.2 \times 10^{-2}$ and $1.5 \times 10^{-2}$, respectively, confirming a correct implementation of the stochastic model of a random mixture of ice particles and air gaps in SCIATRAN.

SZA=68.9°

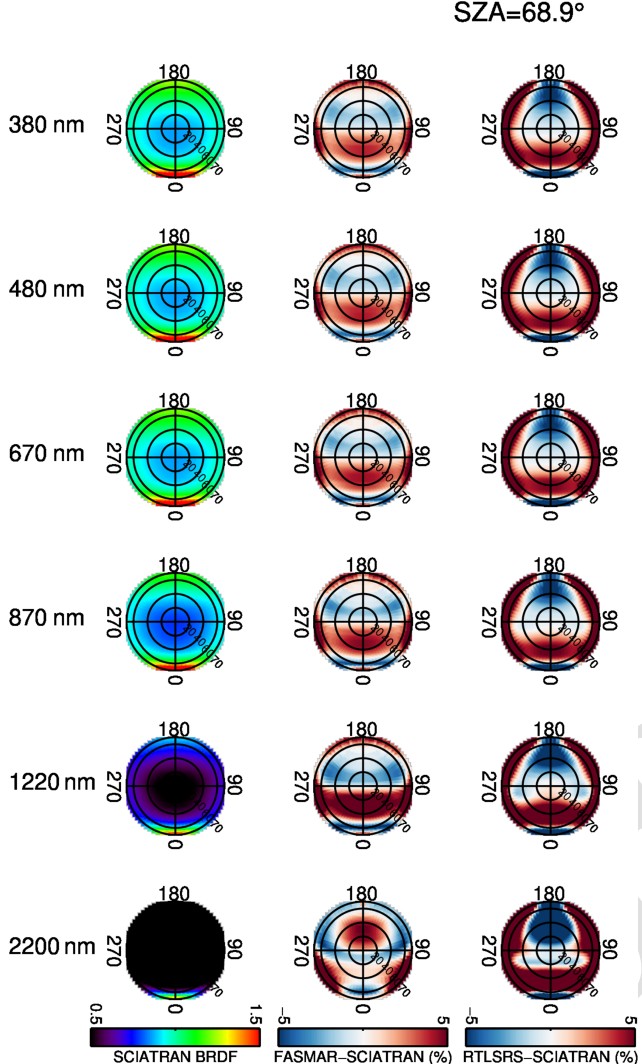

**Figure 5.** BRDF of the snow layer calculated using SCIATRAN and the difference between the results from the FASMAR/RTLSRS models and SCIATRAN simulations for wavelengths of 380, 480, 670, 870, 1220 and 2200 nm and a solar zenith angle of 68.9°.

## 6.4 Melt ponds on sea ice

Melt ponds on sea ice are another surface type typical of high-latitude regions. Comparisons of SCIATRAN simulations for the spectral albedo of light-blue and dark open melt ponds with measurement data are shown in Fig. 7. Measurements of the spectral albedo were performed in the spectral range 300–2500 nm in August 2012 during an R/V *Polarstern* cruise (Malinka et al., 2018). The examples presented in Fig. 7 were selected following Malinka et al. (2018). The spectral albedo of blue and dark melt ponds was measured under cloud-free and cloudy conditions, respectively.

The simulations with SCIATRAN were performed using Eq. (29) of the melt pond reflection described in Sect. 4.3.3

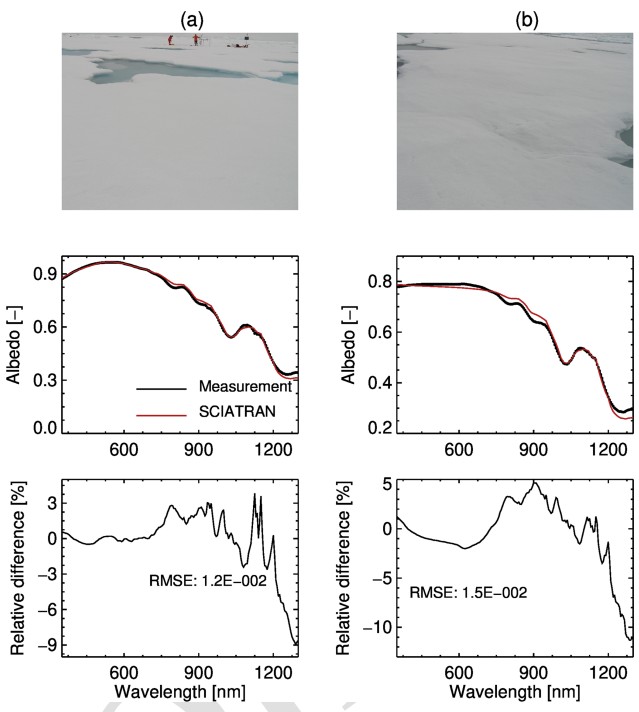

**Figure 6.** Comparisons of measured and simulated bright white ice spectral albedo corresponding to Fig. 8a and c in Malinka et al. (2016). The upper row shows pictures of the measurement conditions, the middle row shows the measured and simulated spectral albedo, and the lower row shows the percentage difference between SCIATRAN simulations and campaign measurements.

and an aerosol-free atmosphere without gaseous absorption. The spectral albedo was calculated as a ratio of the reflected to incident fluxes at the surface. For the blue pond, a cloud-free atmosphere was selected, and the solar zenith angle was set in accordance with Malinka et al. (2018) to 70°. For the dark pond, a water cloud with an optical thickness of 20, a geometrical thickness of 0.5 km and a top height of 1.5 km was considered in addition.

Three parameters required to initialize the BRDF model in SCIATRAN (ice thickness, water depth and transport-scattering coefficient) were obtained by fitting the simulated albedo with respect to the measured one within the spectral range 350–1350 nm. As pointed out by Malinka et al. (2018), measurements outside this spectral range show strong noise contamination. Therefore, they were not used in the fitting process. The following parameters provide the minimum residual: ($H = 281.2$ cm, $z = 14.6$ cm, $\sigma_t = 2.08$ m$^{-1}$) and ($H = 23.7$ cm, $z = 42.8$ cm, $\sigma_t = 0.52$ m$^{-1}$) for the light-blue and dark melt ponds, respectively. It is seen from Fig. 7 that both spectral patterns and magnitudes of the spectral albedo agree well between the SCIATRAN simulations and the measurements, especially for the light-blue melt pond. As expected, the spectral albedo for the dark melt ponds is significantly lower than that for the light-blue ones because

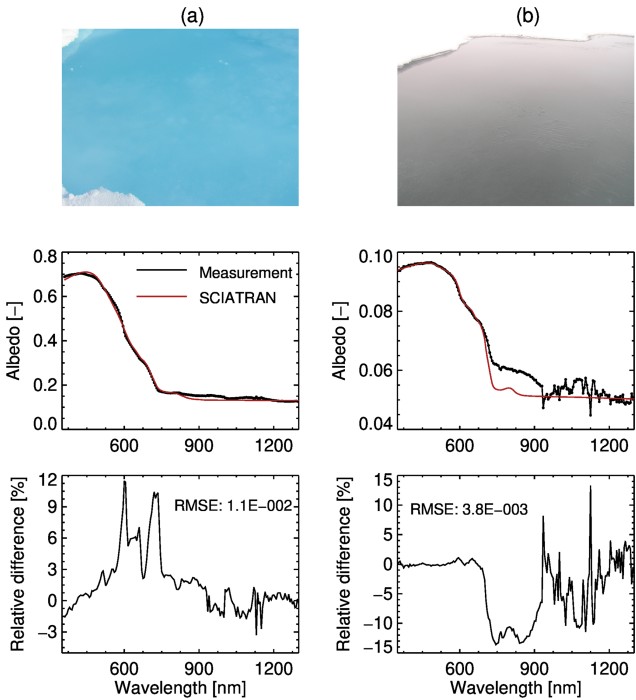

**Figure 7.** Comparisons of the spectral albedo for the light-blue **(a)** and dark **(b)** melt ponds corresponding to Figs. 6a and 8a in Malinka et al. (2018). The upper row shows pictures of the measurement scenes, the middle row shows the measured and simulated spectral albedos, and the lower row shows relative differences between the SCIATRAN simulations and campaign measurements.

of a much larger water depth. Typical spectral features can be seen for both scenes: (1) a higher albedo in the visible range and lower albedo in the infrared range; (2) a weak spectral dependence of the albedo in the NIR range. Both features above can be explained by the wavelength dependence of the water absorption and Fresnel reflection at the air–water interface. Indeed, in contrast to the visible range, water absorbs a significant amount of radiation in the NIR, so that a layer of water a few centimeters thick completely absorbs the radiation. This results in a saturation effect when a variation of the water absorption coefficient does not lead to a variation of the albedo and the wavelength dependence of the albedo is mainly determined by the spectral dependence of the Fresnel reflection.

The RMSE values between SCIATRAN simulations and measurements are $1.1 \times 10^{-2}$ and $3.8 \times 10^{-3}$ for the light-blue and dark melt ponds, respectively, confirming a correct implementation of the ice and melt pond model in SCIATRAN.

# 7 Conclusions

This paper summarizes recent updates and new developments of aerosol, cloud and surface reflectance databases and models in the framework of the software package SCIATRAN. These updates and developments extend the capabilities of the radiative transfer modeling, especially by accounting for vertical inhomogeneities. Vertically inhomogeneous ice and mixed-phase clouds as well as different aerosol types can be easily accounted for within SCIATRAN (V4.6). Additionally, widely used models and databases for the surface BRDF and albedos for different surface types (ocean, land, vegetation, snow and melt pond on sea ice) are available with user-friendly interfaces. We believe that the presented updates and new developments largely extend the application range of the SCIATRAN model in the field of aerosol, cloud and surface remote sensing. The new implementations have been used or are planned for use to support the development and generation of the XBAER global aerosol, cloud and surface data products. Users are advised to use the latest version of SCIATRAN capable of the new features described in this paper. The extended SCIATRAN version is freely available at the website of the IUP, University of Bremen: http://www.iup.physik.uni-bremen.de/sciatran (last access: November 2022).

## Appendix A:  Abbreviations used in this paper

| Abbreviation | Full name |
| --- | --- |
| AATSR | Advanced Along-Track Scanning Radiometer |
| ACLOUD | Arctic Cloud Observations Using airborne measurements during polar Day |
| AERONET | Aerosol Robotic Network |
| AOT | Aerosol optical thickness |
| ART | Asymptotic Radiative Transfer |
| BC | Black carbon |
| BRDF | Bidirectional reflectance distribution function |
| CAS | Cloud, aerosol and surface |
| COT | Cloud optical thickness |
| CCI | Climate Change Initiative |
| C3S | Copernicus Climate Change Service |
| DT | Dark Target |
| DU | DUst |
| EnMAP | Environmental Mapping and Analysis Program |
| EWT | Equivalent water thickness |
| FASMAR | Fast and Accurate Semi-analysical Model of Atmosphere-surface Reflectance |
| FOCAL | Fast atmOspheric traCe gAs retrievaL |
| FMF | Fine-mode fraction |
| GOME | Global Ozone Monitoring Experiment |
| HSI | HyperSpectral Imager |
| IUP | Institute of Environmental Physics |
| LAI | Leaf area index |
| LAM | Leaf mass per area |
| MA | Moderately absorbing |
| MAN | Maritime Aerosol Network |
| MERIS | Medium Resolution Imaging Spectrometer |
| MERRA | Modern-Era Retrospective analysis for Research and Applications |
| MODIS | Moderate Resolution Imaging Spectroradiometer |
| MMR | Mass mixing ratio |
| NIR | Near infrared |
| OC | Organic carbon |
| OLCI | Ocean and Land Colour Instrument |
| OPAC | Optical Properties of Aerosols and Clouds |
| PARASOL | Polarization and Anisotropy of Reflectances for Atmospheric Sciences coupled with Observations from a Lidar |
| PROSAIL | PROSPECT + SAIL |
| POLDER | POLarization and Directionality of the Earth's Reflectances |
| PSD | Particle size distribution |
| RH | Relative humidity |
| RI | Refractive index |
| RPV | Rahman–Pinty–Verstraete |
| RTLSR | RossThick-LiSparseReciprocal |
| RTLSRS | RossThick-LiSparseReciprocal-Snow |
| RMSE | Root-mean-square deviation |
| SA | Strongly absorbing |
| SAVI | Soil-Adjusted Vegetation Index |
| SCIAMACHY | SCanning Imaging Absorption spectroMeter for Atmospheric CHartography |
| SGHM | Snow grain habit mixture |
| SLSTR | Sea and Land Surface Temperature Radiometer |
| SRWSCD | Surface Reflectance Wavelength Shape Constrain Database |
| SS | Sea salt |
| SSA | Single-scattering albedo |
| SU | Sulfate |
| TOA | Top of the atmosphere |
| UV | Ultraviolet |
| VIS | Visible |
| WA | Weakly absorbing |
| WMO | World Meteorological Organization |
| XBAER | eXtensible Bremen Aerosol/cloud and surfacE parameters Retrieval |

## Appendix B: Land surface reflection models

### B1   RPV model including a hotspot correction term

One of the widely used models to approximate the bidirectional reflection function of a land surface is the so-called RPV model proposed by Rahman et al. (1993). In the framework of this model, the surface reflectance is parameterized by the following expressions:

$$R(\mu_v, \mu_s, \varphi) = \rho_0 [\mu_v \mu_s (\mu_v + \mu_s)]^{k-1} F(g) H(G), \quad \text{(B1)}$$

$$F(g) = \frac{1 - g^2}{(1 + 2g \cos\zeta + g^2)^{3/2}}, \quad H(G) = 1 + \frac{1 - \rho_0}{1 + G}, \quad \text{(B2)}$$

$$G = \left[\tan^2\vartheta_v + \tan^2\vartheta_s + 2\tan\vartheta_s \tan\vartheta_v \cos\varphi_r\right]^{1/2}, \quad \text{(B3)}$$

$$\cos\zeta = \mu_v\mu_s - \sqrt{(1 - \mu_v^2)(1 - \mu_s^2)} \cos\varphi_r, \quad \text{(B4)}$$

where $\rho_0$ is the light reflection strength, $k$ is the so-called Minnaert exponent, determining the reflection anisotropy level, and $g$ is the asymmetry parameter controlling the relative amount of forward and backward reflection, $\mu_v = \cos\vartheta_v$ and $\mu_s = \cos\vartheta_s$, with $\vartheta_v$ and $\vartheta_s$ denoting the viewing angle and the solar zenith angle, respectively. The phase angle, $\zeta$, and the scattering angle, $\gamma$, are related as $\zeta = \pi - \gamma$. We note that $\vartheta_v$ and $\vartheta_s$ are defined as angles between the vertical $z$-axis direction and the direction to the satellite and to the Sun, respectively. This means that $\vartheta_v$ and $\vartheta_s$ are always smaller than $\pi/2$; i.e., their cosines, $\mu_v$ and $\mu_s$, are always positive. In contrast to the original publication of Rahman et al. (1993), the relative azimuthal angle, $\varphi_r$, is defined here as zero for the forward scattering, i.e., if the observer looks toward the illumination source. Thus, the relative azimuth angle used by Rahman et al. (1993) differs by $\pi$ from our definition. The function $H(G)$ describes the so-called hotspot effect seen in the measured data as a strong increase in the surface reflectance at $\vartheta_v = \vartheta_s$ and $\varphi_r = \pi$ (see, e.g., Hautecoeur and Leroy, 1998). In SCIATRAN, the RPV model is implemented using the land surface parameters $(\rho_0, k, g)$ presented in Lyapustin (1999), allowing the user to account for the BRDF of various reflecting surfaces such as the forest, savanna and plowed field.

### B2   Kernel-based Ross–Li (RTLSR) model

The kernel-based linear Ross–Li BRDF model is implemented in SCIATRAN according to the following equation:

$$R(\mu_v, \mu_s, \varphi_r) = k_0 + k_v F_v(\mu_v, \mu_s, \varphi_r) + k_g F_g(\mu_v, \mu_s, \varphi_r). \quad \text{(B5)}$$

Here, $F_v(\mu_v, \mu_s, \varphi_r)$ and $F_g(\mu_v, \mu_s, \varphi_r)$ represent the volume-scattering and geometric–optic terms (or kernels), respectively, and $k_0$, $k_g$ and $k_v$ are the input parameters. The geometric–optic kernel accounts for the mutual shadowing of protrusions (Maignan et al., 2004) and is implemented as follows:

$$F_g(\mu_v, \mu_s, \varphi_r) = \frac{\mu_v + \mu_s}{\pi \mu_s \mu_v} (t - \sin t \cos t - \pi)$$
$$+ \frac{1 + \cos\zeta}{2 \mu_s \mu_v},$$
$$\cos t = \frac{2 \mu_s \mu_v}{\mu_s + \mu_v} \sqrt{G^2 + (\tan\vartheta_s \tan\vartheta_v \sin\varphi_r)^2},$$
$$t = \begin{cases} \text{acos } t, & \cos t \leq 1 \\ 0, & \cos t > 1 \end{cases}, \quad \text{(B6)}$$

where $G$ is given by Eq. (B3).

The volume-scattering term approximates the radiative transfer within a vegetation canopy and is calculated in accordance with the Roujean et al. (1992) model as

$$F_v(\mu_v, \mu_s, \varphi_r) = \frac{4 C(\zeta, \zeta_0)}{3 \pi (\mu_v + \mu_s)} \left[\left(\frac{\pi}{2} - \zeta\right) \cos\zeta + \sin\zeta\right] - \frac{1}{3}. \quad \text{(B7)}$$

Here, the phase angle $\zeta$ is given by Eq. (B4), the correction term $C(\zeta, \zeta_0)$ is used to improve the hotspot reflection modeling, and this is implemented following Maignan et al. (2004) as

$$C(\zeta, \zeta_0) = 1 + \left(1 + \frac{\zeta}{\zeta_0}\right)^{-1}, \quad \text{(B8)}$$

where the parameter $\zeta_0$ was found to be between 1 and 2° for most of the reflecting land surfaces (Maignan et al., 2004).

### B3   Kernel-based Ross–Li vegetation–soil model

Following Breon and Vermote (2012), another representation for the kernel-based linear Ross–Li model is implemented in SCIATRAN, which allows one to introduce an alternative wavelength dependence of BRDF:

$$R_\lambda(\mu_v, \mu_s, \varphi_r) = K_0(\lambda) \rho(\mu_v, \mu_s, \varphi_r),$$
$$\rho(\mu_v, \mu_s, \varphi_r) = 1 + K_g F_g(\mu_v, \mu_s, \varphi_r) + K_v F_v(\mu_v, \mu_s, \varphi_r),$$
$$K_0(\lambda) = A_\lambda \left[\frac{2}{\pi} \int_0^1 \int_0^1 \int_0^{2\pi} \rho(\mu_v, \mu_s, \varphi_r) \mu_v \mu_s d\mu_v d\mu_s d\varphi\right]^{-1},$$
$$A_\lambda = c A_{veg}(\lambda) + (1 - c) A_{soil}(\lambda),$$
$$K_v = \alpha_v x + \beta_v, \quad K_g = \alpha_g x + \beta_g,$$
$$x = (A_{\lambda_1} - A_{\lambda_2})/(A_{\lambda_1} + A_{\lambda_2}). \quad \text{(B9)}$$

Here, functions $F_g$ and $F_v$ are given by Eqs. (B6) and (B7), $A_{veg}(\lambda)$ and $A_{soil}(\lambda)$ are the spherical albedos of the vegetation and soil, respectively, $c$ is a fraction of the vegetation, $A_{\lambda_1}$ and $A_{\lambda_2}$ are the spherical albedos of the vegetation–soil mixture at wavelengths $\lambda_1 = 865$ nm and $\lambda_2 = 675$ nm, respectively, and the parameters $\alpha_v$, $\beta_v$ and $\alpha_g$, $\beta_g$ are given in Breon and Vermote (2012). Thus, the linear kernel-based soil–vegetation BRDF in SCIATRAN is initialized with the four input parameters, namely, spectrally dependent Lambertian albedos of the soil, $A_{soil}(\lambda)$, and vegetation, $A_{veg}(\lambda)$, the vegetation fraction, $c$, in the vegetation–soil mixture, and the hotspot parameter $\zeta_0$ (see Eq. B8).

## B4 Modified RPV plus facet model

This BRDF model represents the surface reflectance as a sum of two components, i.e., a volumetric scattering term that is completely depolarizing and a specular reflection term (Diner et al., 2012):

$$\mathbf{R}(\mu_v, \mu_s, \varphi_r) = \mathbf{R}_1(\mu_v, \mu_s, \varphi_r) + \mathbf{R}_2(\mu_v, \mu_s, \varphi_r) . \quad \text{(B10)}$$

The modified Rahman–Pinty–Verstraete function (excluding the hotspot correction) is used for the volumetric term:

$$\mathbf{R}_1(\mu_v, \mu_s, \varphi_r) = \frac{\rho_0}{\pi} [(\mu_v + \mu_s)\mu_v\mu_s]^{k-1} e^{-b\cos\gamma} \, \mathbf{E} , \quad \text{(B11)}$$

where $\rho_0$, $k$ and $b$ are input parameters, $\gamma$ is the scattering angle and the matrix $\mathbf{E}$ is a Mueller matrix with unity in the upper left-hand element and zeros everywhere else.

The second component is polarized, and it represents the facet reflection like the Cox and Munk ocean reflection model:

$$\begin{aligned} \mathbf{R}_2(\mu_v, \mu_s, \varphi_r) = C_2 & \frac{\mathcal{P}(\mu_n) \, S(\mu_v, \mu_s)}{4\mu_v\mu_s\mu_n} \\ & \times \mathbf{L}(-\alpha) \, \mathbf{F}(\gamma, m_r, m_i) \, \mathbf{L}(\alpha_0) , \end{aligned} \quad \text{(B12)}$$

where $C_2$ is a scaling parameter which defines the contribution of this component in the total reflection, $\mathcal{P}(\mu_n)$ is the facet slope probability distribution function, $S(\mu_v, \mu_s)$ is the shadowing factor, $\mathbf{L}(-\alpha)$ and $\mathbf{L}(\alpha_0)$ are rotation matrices and $\mathbf{F}(\gamma, m_r, m_i)$ is the Fresnel reflection matrix depending on the real $m_r$ and imaginary $m_i$ parts of the surface refractive index. The user needs to define at a desired wavelength grid the following model parameters: $\rho_0, k, b$ and $C_2, m_r, m_i$.

## Appendix C: White ice

The analytical expressions for the auxiliary function $T_{\text{dif}}$, average cosine $g$ and absorption coefficient of the yellow substance $\alpha_y$ used in the white ice model described in Sect. 4.3.3 are given below following Malinka (2014, 2015) and Malinka et al. (2016).

The auxiliary function $T_{\text{dif}}$ is defined as

$$\begin{aligned} T_{\text{dif}} = {} & \frac{2(5n^6 + 8n^5 + 6n^4 - 5n^3 - n - 1)}{3(n^3 + n^2 + n + 1)(n^4 - 1)} \\ & + \frac{n^2(n^2 - 1)^2}{(n^2 + 1)^3} \ln\frac{n+1}{n-1} - \frac{8n^4(n^4 + 1)}{(n^4 - 1)^2(n^2 + 1)} \ln n , \end{aligned} \quad \text{(C1)}$$

where $n$ is the refractive index of water. The analytical expression for the average cosine $g$ of the phase function is given by

$$g = \frac{1}{\omega} \left[ r_1 + \frac{n^2 t_1^2}{T_{\text{dif}}(1 - n^2) - r_1 + n^4(1 + \alpha a)} \right] , \quad \text{(C2)}$$

where $r_1$ and $t_1$ are

$$\begin{aligned} r_1 = {} & \frac{\begin{aligned}n(3n^{11} + 3n^{10} + 25n^9 + 25n^8 + 22n^7 - 282n^6 + 138n^5 \\ + 186n^4 + 151n^3 - 89n^2 + 13n - 3)\end{aligned}}{24(n+1)(n^4 - 1)(n^2 + 1)^2} \\ & + \frac{8n^4(n^6 - 3n^4 + n^2 - 1)}{(n^4 - 1)^2(n^2 + 1)^2} \ln n \\ & - \frac{(n^8 + 12n^6 + 54n^4 - 4n^2 + 1)(n^2 - 1)^2}{16(n^2 + 1)^4} \ln\frac{n+1}{n-1} , \end{aligned}$$

$$\begin{aligned} t_1 = {} & \frac{3n^8 + 3n^7 - 17n^6 + 55n^5 - 39n^4 - 7n^3 - 27n^2 - 11n - 8}{24(n+1)(n^4 - 1)n} \\ & - \frac{(n^2 - 1)^4}{16(n^2 + 1)^2 n} \ln\frac{n+1}{n-1} + \frac{4n^5}{(n^4 - 1)^2} \ln n . \end{aligned} \quad \text{(C3)}$$

The absorption coefficient of the yellow substance is given by

$$\alpha_y(\lambda) = \begin{cases} \alpha_y(\lambda_0) \, e^{-0.015\,(\lambda - \lambda_0)} , \\ \quad \lambda \leq 500\,\text{nm} , \\ \alpha_y(\lambda_0) \, e^{-0.015\,(500 - \lambda_0) - 0.011(\lambda - 500)} , \\ \quad \lambda > 500\,\text{nm} , \end{cases} \quad \text{(C4)}$$

where $\lambda_0 = 390$ nm, and $\alpha_y(\lambda_0)$ is the absorption coefficient of the yellow substance at the wavelength $\lambda_0$.

## Appendix D: Melt ponds

The albedo of the under-pond ice layer needed by the melt pond model described in Sect. 4.3.3 is calculated according to the two-stream approximation developed by Zege et al. (1991):

$$A_b = A_0 \frac{1 - e^{-2\gamma\tau}}{1 - A_0^2 e^{-2\gamma\tau}} , \quad \text{(D1)}$$

where $A_0 = 1 + t - \sqrt{t\,(t+2)}$ is the albedo of the semi-infinite layer with the same optical characteristics, $t = 8\alpha_i/3\sigma_t$, $\tau = (\sigma_t + \alpha_i)\,H$ is the layer optical thickness, $H$ is its geometrical thickness and $\gamma$ is the asymptotic attenuation coefficient given by

$$\gamma = \frac{3}{4} \frac{\sigma_t}{\sigma_t + \alpha_i} \sqrt{t\,(t+2)} . \quad \text{(D2)}$$

Here, $\alpha_i$ and $\sigma_t$ are the ice absorption coefficient and transport-scattering coefficient of ice (see, e.g., Chandrasekhar, 1950), respectively.

*Code and data availability.* The current version of SCIATRAN is available from the institution's website: http://www.iup.physik. uni-bremen.de/sciatran (last access: November 2022) under LGPL licence. The exact version of the model and input data used to produce the results used in this paper is archived on Zenodo (https://doi.org/10.5281/zenodo.7376666, Rozanov et al., 2022).

*Author contributions.* LM and VR designed the experiments, and LM, VR and AR developed the model code and performed the simulations. LM and VR prepared the manuscript with contributions from all the co-authors. JPB provided general oversight and guidance.

*Competing interests.* The contact author has declared that none of the authors has any competing interests.

*Acknowledgements.* The valuable discussions with Robert Levy, Peter Colarco, Ping Yang, Bryan A. Baum, Masanori Saito, Oleg Dubovik and Aleksey Malinka are highly appreciated.

*Financial support.* This research has been supported by the Deutsche Forschungsgemeinschaft (grant no. 268020496 – TRR172).

The article processing charges for this open-access publication were covered by the University of Bremen.

*Review statement.* This paper was edited by Slimane Bekki and reviewed by two anonymous referees.

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

**Remarks from the typesetter**

TS1    Please give an explanation of why this needs to be changed. We have to ask the handling editor for approval. Thanks.

TS2    Please give an explanation of why this needs to be changed. We have to ask the handling editor for approval. Thanks.

TS3    Please give an explanation of why this needs to be changed. We have to ask the handling editor for approval. Thanks.

TS4    Please give an explanation of why this needs to be changed. We have to ask the handling editor for approval. Thanks.

TS5    Please give an explanation of why this needs to be changed. We have to ask the handling editor for approval. Thanks.

TS6    Please give an explanation of why this needs to be changed. We have to ask the handling editor for approval. Thanks.

TS7    Please give an explanation of why this needs to be changed. We have to ask the handling editor for approval. Thanks.

TS8    Please give an explanation of why this needs to be changed. We have to ask the handling editor for approval. Thanks.

TS9    Please give an explanation of why this needs to be changed. We have to ask the handling editor for approval. Thanks.