# Peer review of "SCIATRAN software package (V4.6): update and further development of aerosol, clouds, surface reflectance databases and models"

_Geoscientific Model Development, 2022_

## Author Comment (AC1)

Dear Editor, dear reviewer,

Thanks for the valuable comments, which help to improve the quality of the paper. The detailed replies are addressed below point by point in blue.

Best regards,

Linlu Mei on behalf of all co-authors

**GENERAL COMMENTS**

The article aims at describing the improvements of the cloud and aerosol database of SCIATRAN software package. A precise description of the optical properties of cloud and aerosol is an important part of a radiative transfer package. The article gives a good overview of this database. The new features are well distinguished. The article presents a scientific interest and is worth being published in GMD. However, some points could be improved:

- To avoid the need to refer to external articles, it would have been preferable to give some more details on the components of CAS, which have not been updated. Of course, if wouldn't be necessary to delve into the details, since they are available in the references.

Response: The components of CAS which have not been updated include

Cloud:

- liquid water clouds with effective radius of water droplets in the range 4 20 µm
- ice clouds consisting of ice crystals having the shape of second generation fractal

Aerosol:

- > OPAC version 3.0
- ➤ WMO
- > LOWTRAN

Surface:

> RPV

**modified RPV models**

RTLSR model

An extended description for the above listed components is included in the revised version (See Appendix B in the revised version).

- It would have been interesting to have more comparisons of modelizations done with the new version of SCIATRAN/CAS with former versions and with other modelizations or measurements.

Response: Some comparisons for the updated and old versions of CAS databases are included in the revised version of the manuscript. In particular, the comparison between OPAC version 3 (former version) and OPAC version 4 (new version) databases.

- Even if the acronyms are properly defined throughout the text, it seems to me that it would be more easy for the reader if they were gathered in an appendix.

Response: A list of acronyms is included in the revised version (See Appendix A in the revised version).

**SPECIFIC COMMENTS**

Lines 41 to 52 about the use of SCIATRAN do not seem necessary to me, as well. The same applies to figure 1.

Response: In the revised manuscript, we have removed Fig. 2 and rewritten this section to provide only the fundamental-needed information about SCIATRAN applications. As to Figure 1, we would prefer (if it is also OK from reviewer side) to keep it since this is the first overview figure of SCIATRAN users after 33 years developments. We believe Figure 1 will still provide information especially for new users to get more confidence to join the SCIATRAN community.

Lines 94 and 95, it is said that the calculation can be performed 175.44 nm to 40 micrometer.

Response: Different models and databases are valid in different spectral ranges. In the spectral range 225 – 2500 nm most of the described databases can be used. For gaseous absorbers, Rayleigh scattering and thermal emission, the calculations can be performed in the spectral range 175.44 nm – 40  $\mu$ m. We have updated the description in the revised version (see lines 66-69 in the revised version)

Lines 99 and 100, it is mentioned that aerosol and cloud scattering can be taken into account.

Response: Yes, aerosol and cloud scattering could be considered in all previous SCIATRAN versions. However, in V4.6, we have largely extended available databases for aerosol and clouds.

However, line 180 and 181, it is said that database can be used "at least" between 225 nm and 2.5 micrometer. Is seems to me that there is a contradiction there: is it possible to model aerosols and clouds between 175.44 and 225 nm, and between 2.5 and 40 micrometer or not? This point should be made clear in the article.

Response: Different models and databases are valid in different spectral ranges. In the spectral range 225 – 2500 nm most of the described databases can be used. For gaseous absorbers, Rayleigh scattering and thermal emission, the calculations can be performed in the spectral range 175.44 nm – 40  $\mu$ m. We have updated the description in the revised version (see lines 66-69 in the revised version)

Line 111, it would be interesting to specify with jacobian can be computed (derivative of radiances with respect to which variables ?)

Response: For CAS related parameters Jacobians can be calculated for the cloud optical thickness, cloud top and bottom heights, effective radius of water droplets and ice crystals, aerosol number density, surface albedo, etc. The complete list of atmospheric and surface parameters, for which the Jacobians can be calculated, can be found in the user guide. This information is also included in the revised manuscript (see lines 88-90 in the revised version).

Line 118, I understand that the goal of the paper is not to delve into details which have been extensively covered in other papers, but the list of the solvers which can be used should be precised.

Response: The information about solvers is included in the revised manuscript (see lines 97-102 in the revised version).

Line 187: by "optical characteristics" do you mean refractive indexes or (SSA, extinction coefficient and phase function?)

Response: Optical characteristics are referring to the extinction coefficient, single scattering albedo and scattering matrix (or phase function in the scalar case).

We have clarified it in the revised manuscript (see lines 176-177 in the revised version).

Line 205: it could be interesting to give references for OPAC version 3. This could also be done in section 4.1.1, event if the details have been given in [Rozanov, 2014]

Response: Reference for OPAC version 3 (Hess et al., 1998) is added in the revised manuscript.

M. Hess, P. Koepke, and I. Schult. Optical properties of aerosols and clouds: The software package OPAC. Bulletin of the American Meteorological Socie, 79:831 - 844, 1998

Section 4.1.1 : it could be interesting to give references for OPAC version 3. This could also be done in line 205, even if the details have been given in [Rozanov, 2014]

Response: Reference for OPAC version 3 (Hess et al., 1998) is added in the revised manuscript (see above)

Line 205-212: I understand that the refractive indexes and size distributions from OPAC are used to compute the topical properties of an aerosol type. Why not use directly optical properties (i.e. SSA, extinction coefficients and phase function) from OPAC?

Response: There are two reasons:

i) the database containing refractive indices and size distributions is significantly more compact in contrast to the database containing SSA, extinction coefficients and phase function;

ii) the OPAC database does not provide all elements of scattering matrix. As described in Hess et al (1998), to avoid repeated Mie- or ray-tracing calculations, optical properties of basic aerosol components and clouds are stored in OPAC, the optical properties are the extinction coefficient, the scattering coefficient, the absorption coefficient, the volume phase function, the single scattering albedo and the asymmetry parameter.

Line 218-225: it is mentioned above that the optical properties are computed from the refractive indexes. Here, it is said that they can be set for different altitudes. Does it mean that optical properties can either be directly specified or computed from refractive indexes ?Even if OPAC v3 is not the focus of this paper, the way optical properties are defined (specified or computed from refractive indexes) should be precised.

Response: In the case of OPAC V3.0, the aerosol optical properties are computed using refractive indexes and size distributions from the database. The calculations are

performed by SCIATRAN automatically employing the incorporated Mie code. SCIATRAN users can select different OPAC components in different altitude layers.

More detail on using OPAC is included in the revised manuscript (see lines 202-204 in the revised version).

Line 243: the same applies to OPAC aerosols, where optical properties can be specified.

Response: In the case of OPAC V4.0, the aerosol optical properties are specified using SSA, extinction coefficients and phase functions from the database.

Line 243 refers to OPAC V 4.0 We made it more clearly in the revised manuscript (see line 236 in the revised version).

Line 265: logarithms/exponentials of dimensioned quantities units are used. The formula should be modified so that no dimensioned quantity is used as an argument to ln or exp.

Response: The argument of exponential in Eq.1 is dimensionless because ln r - ln  $r_v = ln(r/r_v)$  and standard deviation is defined with respect to the dimensionless variable ln(r/rv).

Line 286 Same remark. Moreover, the dimension of both sides should be the same.

Response: The dimension of both sides of Eq. 3 is the same. We note that N0 is the number of aerosol particles in a vertical column of unit area, having dimension  $1/\mu m^2$ , V0 is the total volume of particles in vertical column of unit area, having dimension  $\mu m^3/\mu m^2$  (see Table 2 of manuscript), therefore the dimension of the right hand side of Eq. 3 is  $1/\mu m^2$ , i.e., the same as dimension of left-hand side.

Line 296: it is said that the user can define the shape of the aerosol number density vertical distribution. What is the default shape (i.e. if the users does not define it)?

Response: The default shape is a constant value within an aerosol layer.

We have added this information in the revised manuscript (see lines 287-288 in the revised version).

Paragraph 4.1.4 : why a "dust aerosol type" is mentioned, whereas, in this paragraph, other aerosol types are mentioned (OPAC, XBAER-OC, MODIS-DT), which are considered in other paragraphs ? This paragraph should focus on the Dubovik dust aerosol type, which is not described elsewhere.

Response: Section 4.1.4 describe Dubovik dust aerosol type, we have changed the title of this sub-section from 'Dust aerosol type' to 'Dubovik dust aerosol type' in the revised manuscript.

Line 389: for the "size bins" tracer: is it an aerosol with a defined size ? This aerosol type should be precised.

Response: Yes, in MERRA, each aerosol component is defined by a fixed size distribution. We have added this information in the revised manuscript (see lines 279-281 in the revised version).

Line 406: mixing ratio: with respect to dry air?

Response: mass-mixing ratio is given for dry air.

Line 442: is soot accounted for by adding an extra aerosol type? Or by considering a layered sphere with both water (liquid or solid) and soot?

Response: Contribution of soot is accounted for by adding a specific aerosol type.

Line 490: in [Yang, 2013], there are 11 habits, not 9.

Response: This has been updated in the revised manuscript (see lines 471-475 in the revised version).

Line 497: It would be nice to have more details about the implementation, even if the full description is available in [Pohl 2020]

Response: The extinction cross-section, single scattering albedo, non-zero elements of the scattering matrix, effective radius, projected area, and volume of each habit at 189 dimensions were used from original Yang database. The expansion coefficients were calculated expanding the elements of the scattering matrix in the generalised spherical functions.

We have included additional details with respect to the implementation of Yang database in the revised manuscript (see lines 478-483 in the revised version).

Line 565 and 580: without going into the details of Malinka's articles, it would be nice to recall briefly the principle of stereological approach and the definition of chord length.

Response: A brief description of the main principle of stereological approach and the definition of chord length are included in the revised manuscript (see lines 545-553 in the revised version).

Lines 588 and after: some details would be given about the RPV and Kernel-based Ross-Li. Not a full description, but something basic which would avoid the need to look in the references.

Response: A brief description of the above-mentioned models is given in Appendix B of the revised manuscript.

Line 775: "newly implemented in SCIATRAN aerosol types" => "aerosol types newly implemented in SCIATRAN".

Response: This has been corrected in the revised manuscript.

Figure 3b: it is difficult to see the difference between the forward peaks of the different aerosols, which are mentioned in lines 800-810. Perhaps adding another figure for the phase function, zooming on 0 degree scattering angle.

Response: A sub-panel inside Fig 3b is included to zoom-in the region from 0 to 15 degree scattering angle.

---

## Author Comment (AC2)

Dear Editor, dear reviewer,

Thanks for the valuable comments, which help to improve the quality of the paper. The detailed replies are addressed below point by point in blue.

Best regards,

Linlu Mei on behalf of all co-authors

The paper describes the latest updates and additions to the already extensive suite of aerosol, cloud and surface reflectance databases implemented in SCIATRAN. The new databases are adopted from mostly recent work of, and published in the peer-reviewed literature by, a number of researchers. The paper is fairly well written and accessible. It should provide a useful reference to current and future users of SCIATRAN.

I understand the goal is to describe the updated databases, but I think a high-level description of the state of SCIATRAN would be useful in addition to the references that are already provided. This should include the radiative transfer solver(s) implemented, databases that existed before the recent update, as well as a reasoning for replacing them, if any.

Response: An extended description for the above-mentioned aspects is included in the revised version. For example, radiative transfer solvers (see lines 97-102 in the revised version); existing data and models (see Appendix B in the revised version). More information can also be found at lines 132 – 137 in the revised version.

It is not entirely clear to me whether the new aerosol databases are in addition to the "old" aerosol types or the they replace the "old" ones.

Response: The CAS databases and models from the previous version of SCIATRAN are still available in V4.6.

We have included additional clarification on this topic (see lines 132 – 137 in the revised version).

The databases implemented in SCITRAN is now quite extensive. While reading the manuscript I was hoping to find some sort of guidance or recommendation as to which database should be used under various situations. I don't mean the trivial scenarios like land aerosol models over land and oceanic aerosol models over ocean, or water and ice clouds depending on what the user want to simulate. I am thinking of

especially of the bidirectional surface reflectance databases. Which one would be the most appropriate to use when all inputs required for the models are available?

Response: When all input parameters required for a selected surface type are available, we can give the following recommendations:

For an ocean surface, users have only one option to use

For a snow-free land surface, the following choices can be made:

1) For users preferring a simple globally applicable BRDF model, the XBAER BRDF model is recommended. In this model only one parameter (Soil-Adjusted Vegetation Index) is needed as the input.

2) For users interested is specific topics related to the vegetation properties, the PROSAIL is recommended.

3) For users working with multi-angle observations and only interested in a non-polarized modelling, the Ross-Li model is recommended

4) For users working with multi-angle observations and interested in a polarized modelling, a combination of the RPV and Facet models is recommended.

For a snow-covered land surface, FASMAR is recommended.

We have included the above information in the revised manuscript (see lines 585 – 598 in the revised version)

SPECIFIC COMMENTS:

Lines 113-14: Are the fluxes only for selected wavelengths, can the user request spectrally integrated quantities too?

Response: The calculation of spectrally integrated quantities is available. Different spectral response functions can be selected by user.

Lines: 126-128: I assume this means the gas absorption calculations use the HITRAN 2020 database. I understand the discussion of it is out of scope, but please at least provide a reference.

Response: The HITRAN 2020 reference is included in the revised manuscript.

Gordon, I. E., Rothman, L. S., Hargreaves, R. J., Hashemi, R., Karlovets, E. V., Skinner, F. M., Conway, E. K., Hill, C., Kochanov, R. V., Tan, Y., Wcisło, P., Finenko, A. A., Nelson, K., Bernath, P. F., Birk, M., Boudon, V., Campargue, A., Chance, K. V., Coustenis, A., Drouin, B. J., Flaud, J.-M., Gamache, R. R., Hodges, J. T., Jacquemart,

D., Mlawer, E. J., Nikitin, A. V., Perevalov, V. I., Rotger, M., Tennyson, J., Toon, G. C., Tran, H., Tyuterev, V. G., Adkins, E. M., Baker, A., Barbe, A., Canè, E., Császár, A. G., Dudaryonok, A., Egorov, O., Fleisher, A. J., Fleurbaey, H., Foltynowicz, A., Furtenbacher, T., Harrison, J. J., Hartmann, J.-M., Horneman, V.-M., Huang, X., Karman, T., Karns, J., Kassi, S., Kleiner, I., Kofman, V., Kwabia-Tchana, F., Lavrentieva, N. N., Lee, T. J., Long, D. A., Lukashevskaya, A. A., Lyulin, O. M., Makhnev, V. Yu., Matt, W., Massie, S. T., Melosso, M., Mikhailenko, S. N., Mondelain, D., Müller, H. S. P., Naumenko, O. V., Perrin, A., Polyansky, O. L., Raddaoui, E., Raston, P. L., Reed, Z. D., Rey, M., Richard, C., Tóbiás, R., Sadiek, I., Schwenke, D. W., Starikova, E., Sung, K., Tamassia, F., Tashkun, S. A., Vander Auwera, J., Vasilenko, I. A., Vigasin, A. A., Villanueva, G. L., Vispoel, B., Wagner, G., Yachmenev, A., and Yurchenko, S. N.: The HITRAN2020 molecular spectroscopic database, J. Quant. Spectrosc. Radiat. Transf., 277, 107949, https://doi.org/10.1016/j.jqsrt.2021.107949, 2022.

Lines 259-262: It is said that the continental aerosol type is not included in the SCIATRAN database because according to the authors' investigation it is not used in the XBAER algorithm for MERIS. It may be fine since the continental aerosol type is generally regarded "old" by some researchers but I'd think there should be a more substantial justification for excluding it beyond the fact that the XBAER algorithm does not use it for a specific instrument.

Response: Yes, the continental aerosol type is not included in SCIATRAN. One of the important reasons is, that the new implementations in SCIATRAN are driven mostly by the further development of the XBAER algorithm. In the XBAER algorithm, we are not using the continental aerosol type. Even in the NASA MODIS Dark-Target algorithm, continental aerosol type is only used as a back-up type when weakly-, moderately- and strongly absorbing aerosol types do not fit in the retrieval. Furthermore, as the reviewer mentioned, this continental aerosol type is not new.

We have extended the explanation in the revised manuscript (see lines 255 – 260 in the revised version).

Lines: 445-449: In the previous version, the effective radius of water droplets was between 4 and 20 microns. So, do you mean the database was extended to include the range 2 - 4 microns, or that the "old" database was replaced with a new one having effective radius between 2 and 40 microns?

Response: Database for water droplet sizes between 2 – 4 µm and 20 – 40 µm was calculated and added to the original 'old' database.

Table 3 caption: I think the correct URL is https://darktarget.gsfc.nasa.gov/atbd/ocean-algorithm

Response: Both the link above and in the paper contain the ocean aerosol models

Line 459: Is there a reference for the relationship between mode radius and effective radius?

Response: See, e.g., Kokhanovsky A.A. Cloud optics. Dordrecht: Springer; 2006.

At places where implementation of databases is discussed the choice of the specific way a particular database was implemented could be discussed. For example, re the ice particle habit (lines 520-525), has the linear interpolation been suggested by the database developer? I am not saying that a linear interpolation is not adequate; I am only suggesting that when it is appropriate the choice should be justified as much as possible, especially in those cases when multiple choices are possible.

Response: The linear interpolation with respect to the logarithm of maximal dimension of grain habits was implemented into the SCIATRAN software according to the suggestion of Saito et al., see

M. Saito, P. Yang, N. G. Loeb and S. Kato "A novel parameterization of snow albedo based on a two-layer snow model with a mixture of grain habits", J. Atmos. Sci. 2019, v76, 1419-1436.

Another interpolation type cannot be selected by user. We have included this information in the revised manuscript (see line 506 in the revised version).

Line 577: Are "typical values" of the mean chord length and optical thickness of snow provided in the code or database for a broad category of snow, say fresh and old snow? I assume users could read the Malinka (2014) paper, but still, having such values listed in the paper would be useful for casual users who just want to see the sensitivity to these parameters without prescribing unrealistic values.

Response: Typical values of chord length and optical thickness for different snow types are given in the revised version of manuscript (see Table 6 of the revised manuscript).

Line 808: The Dubovik and OPAC 4.0 dust models also have relatively large differences in the backward scattering direction (scattering angle 150-180 degrees).

Response: Corrected.

Lines 903-911: I am not sure I understand this part. What is being shown on the vertical axis in the lower panel of Fig. 5? Is it the reflectance from a single aerosol component divided by the reflectance from all 5 components, or the reflectance of 4

components (excluding one aerosol type) divided by the reflectance from all 5 aerosol types?

Response: We have updated Fig.5, especially the lower panel. In the revised version, the lower panel shows the TOA reflectance simulated considering individual aerosol components.

Line 2021: I assume the authors mean user-friendly interfaces. Is it really necessary to say it since it has not been shown in the paper?

Response: We believe users can get very quickly a feeling how to make correct settings once they start using SCIATRAN. Such a statement can also 'encourage' potential users (especially who are afraid of a complexity of the software) to use SCIATRAN. Thus, we would like to keep it in the revised manuscript.

TECHNICAL CORRECTIONS (could be more):

Line 145: "the explore of" should read " the exploration of".

Response: Done

Line 152: "of Young database" should read "**the** Young database".

Response: Done

Lines 191-192: "... optical properties those six components" should read "... optical properties **of** those six components".

Response: Done

Line 249: "Virible" should read "Visible".

Response: Done

Line 311: "wavelegths" should read "wavelengths".

Response: Done

Line 351: "Many investigations has …" should read "Many investigations have …"

Response: Done

Line 548: "Combing" should read "Combining".

Response: Done

Line 588: Perhaps "build-in" should read "built-in"?

Response: Done

Lines 773-774: Would read better "... the new aerosol types implemented in SCIATRAN" or "... the aerosol types newly implemented in SCIATRAN".

Response: Done

Line 807: "paramterization" should read "parameterization".

Response: Done

Figure 3 caption: See comment above for lines 773-774.

Response: Done

Line 824: "show layer" should read "snow layer".

Response: Done

Lines 828-829: Instead of "the scattering processes plays very important role" the authors could write "the scattering process plays a very important role" or "scattering processes play a very important role".

Response: Done

Lines 855-856: "responce" should read "response"; "instrument" should read "instrument".

Response: Done

Line 859-860: "to the time of writing" should read "at the time of writing".

Response: This sentence is deleted because the proposal has been accepted and this sentence is not needed any more.

Line 927: "chanels" should read "channels".

Response: Done

Lines 983-984: "with the optical thickness of 20" should read "with an optical thickness of 20"; "the geometrical thickness" should read "a geometrical thickness"; "and the top height" should read "and a top height".

Response: Done

---

## Author Comment (AC3)

Dear Dr. Juan A. Añel,

Thanks for the valuable comment, which helps to improve the way to release the software to users. The detailed reply is addressed below in blue.

Best regards,

Linlu Mei on behalf of all co-authors

Dear authors,

Unfortunately, after checking your manuscript, it has come to our attention that it does not comply with our "Code and Data Policy".

https://www.geoscientific-model-development.net/policies/code_and_data_policy.html

You have archived your code on a server at the University of Bremen. However, this is not a suitable repository. You must use other alternatives for long-term archival and publishing (check those listed in our policy). Therefore, please, publish your code in one of the appropriate repositories, and reply to this comment with the relevant information (link and DOI) as soon as possible, as it should be available for the Discussions stage. Also, please, include the relevant primary input/output data. In this way, you must include the modified 'Code and Data Availability' section in a potential reviewed version of your manuscript, the DOI of the code (and another DOI for the dataset if necessary).

Please, note that currently, on the web page of the University of Bremen, it reads that it is necessary to register to get access to the code, and this is against the policy of our journal. Therefore, when uploading the code to the new repository, you must make it available to anyone without registering, logging in, etc.

Be aware that failing to comply with this request will result in the rejection of your manuscript for publication.

Regards,

Juan A. Añel
Geosci. Model Dev. Exec. Editor
**Citation**: https://doi.org/10.5194/gmd-2022-153-CEC1

Response: The current version of SCIATRAN is available from the institution website: https://www.iup.uni-bremen.de/sciatran/ under the LGPL licence.

The exact version of the model and input data used to produce the results used in this paper is archived on Zenodo (10.5281/zenodo.7376666).